# Rapid expansion and visual specialisation of learning and memory centres in the brains of Heliconiini butterflies

Antoine Couto[1,2,7], Fletcher J. Young[1,2,3,7], Daniele Atzeni[1,4], Simon Marty[2,5], Lina Melo-Flórez [3], Laura Hebberecht[1,2,3], Monica Monllor[3], Chris Neal[6], Francesco Cicconardi[1], W. Owen McMillan[3] & Stephen H. Montgomery [1,3] ✉

Changes in the abundance and diversity of neural cell types, and their connectivity, shape brain composition and provide the substrate for behavioral evolution. Although investment in sensory brain regions is understood to be largely driven by the relative ecological importance of particular sensory modalities, how selective pressures impact the elaboration of integrative brain centers has been more difficult to pinpoint. Here, we provide evidence of extensive, mosaic expansion of an integration brain center among closely related species, which is not explained by changes in sites of primary sensory input. By building new datasets of neural traits among a tribe of diverse Neotropical butterflies, the Heliconiini, we detected several major evolutionary expansions of the mushroom bodies, central brain structures pivotal for insect learning and memory. The genus *Heliconius*, which exhibits a unique dietary innovation, pollen-feeding, and derived foraging behaviors reliant on spatial memory, shows the most extreme enlargement. This expansion is primarily associated with increased visual processing areas and coincides with increased precision of visual processing, and enhanced long term memory. These results demonstrate that selection for behavioral innovation and enhanced cognitive ability occurred through expansion and localized specialization in integrative brain centers.

The diversity of animal behaviours, senses and cognitive abilities is the result of evolutionary innovations and refinements in neural systems. Throughout animal evolution, the nervous system has acquired new features, built on previously existing functions, which have allowed animals to evolve new ways of perceiving and interacting with their environment. Models of brain evolution increasingly emphasise the coordinated evolution of functionally related networks in this process, but with localised specialisation where selection targets refinement of existing functions[1–3]. Under this model, increased behavioural precision, or diversification, can occur through replication of established cell types and circuits within generally conserved networks[4,5]. While existing data provide patterns of variation consistent with this process, demonstrating associations between specialised refinements of specific neuronal circuits and the evolution of novel behaviours remains challenging.

Cases of ecological innovation can provide unique opportunities to link neural and behavioural evolution[6,7] particularly when

[1]School of Biological Sciences, University of Bristol, Bristol, UK. [2]Department of Zoology, University of Cambridge, Cambridge, UK. [3]Smithsonian Tropical Research Institute, Gamboa, Panama. [4]Department of Life Science, University of Trieste, Trieste, Italy. [5]École Normale Supérieure de Lyon, Université Claude Bernard Lyon 1, Université de Lyon, Lyon, France. [6]Wolfson Bioimaging Facility, University of Bristol, Bristol, UK. [7]These authors contributed equally: Antoine Couto, Fletcher J Young. ✉e-mail: s.montgomery@bristol.ac.uk

grounded in robust phylogenetic frameworks. For example, uniquely among butterflies, adult *Heliconius* actively collect and digest pollen[8,9], providing an adult source of essential amino acids[8,10] and facilitating a greatly extended reproductive lifespan[11]. This dietary innovation is accompanied by the evolution of trap-line foraging, where individuals learn foraging routes between resources with high spatial and temporal fidelity[12–14]. Among insects, this foraging strategy is also found among some species of bee, and requires the ability to form vector memories[15,16] and store large amounts of visual scenes[17,18], potentially including landmark cues[19,20]. This suggests that the evolution of pollen feeding in *Heliconius* may have required neural and cognitive enhancements to support optimal foraging for low-density pollen resources through enhanced spatial memory. Indeed, preliminary evidence suggests that *Heliconius* have expanded mushroom bodies, an insect learning and memory centre[21], relative to their non-pollen-feeding relatives in the Heliconiini tribe[22,23]. While mushroom bodies have previously been viewed as non-essential for spatial memory in *Drosophila*[24,25], more recent data do suggest a role in spatial memory[26] with more visual input to the calyx than previously appreciated[27]. Combined with mounting evidence from empirical[28–32] and theoretical modelling[33,34] in other insects, these data strongly implicate the mushroom bodies in learnt spatial behaviours. However, the adaptive benefit of mushroom body expansion remains largely unestablished. Increased mushroom body size may facilitate increased memory space, which is likely essential for the memorisation of multiple visual scenes to support learned foraging routes across large spatial scales[33]. However, given their role in sensory integration and both elemental and more complex learning tasks[35,36] larger mushroom bodies may also support more general cognitive enhancements through greater sensory discrimination and behavioural precision, through sparse coding of stimuli[37,38]. Here, we provide an extensive analysis of mushroom body expansion in *Heliconius*, with new, phylogenetically dense data across all major Heliconiini lineages. By incorporating multiple neuroanatomical and behavioural traits, we establish a rich system to link neural elaboration and behavioural innovation.

## Results and discussion

### Extensive variability in Heliconiini mushroom body size

We generated an extensive dataset of brain composition for 318 wild-caught individuals from 41 Heliconiini species (average 8 individuals/species), including 30 species and sub-species of *Heliconius* and representatives from each Heliconiini genus (Fig. 1). 3D volumetric reconstructions of whole brains stained with a synaptic marker revealed a high degree of variation in mushroom body size across the tribe. In raw volume, mushroom bodies vary by 26.5-fold, from ~3 × 10⁶ μm³ to ~70 × 10⁶ μm³ per hemisphere (Supplementary Data 1 and 2). This level of variation is unmatched by major visual or olfactory brain structures (medulla: 7.9-fold variation; antennal lobe: 6.2-fold variation), or the remaining volume of the central brain (7.8-fold variation). Across all individuals, correcting for allometric (general size variation) effects and phylogenetic relatedness using *MCMCglm*, mushroom body volume is positively associated with variation in both visual and olfactory structures (medulla: $p_{MCMC} = 0.002$; antennal lobe: $p_{MCMC} < 0.001$). However, significant variation between phylogenetic groups still persists, with *Heliconius* having significantly larger mushroom bodies compared to other genera ($p_{MCMC} < 0.001$). This is not the case for major visual or olfactory brain regions (medulla: $p_{MCMC} = 0.482$; antennal lobe: $p_{MCMC} = 0.202$; Fig. 2E–H). Hence, a major portion of interspecific differences in mushroom body size is independent of variation in sensory brain regions. Repeating this analysis including more narrow phylogenetic groupings (outgroup genus, or *Heliconius* subclades) suggests that variation in mushroom body size is not distributed bimodally, rejecting a simple shift between *Heliconius* and other genera, and instead varies both within *Heliconius* and across the outgroup genera (Supplementary Note 1, Fig. S1 and Tables S1 and 2). Interestingly, we also identify a *Heliconius* specific effect of sex (interaction *Heliconius**Sex $p_{MCMC} < 0.001$), with females having larger mushroom bodies on average than males ($p_{MCMC} < 0.001$; Supplementary Note 2 and Fig. S2). Among wild-caught *Heliconius* females tend to exhibit larger pollen loads[39], forage earlier in the day and cover smaller areas compared to males, focusing on more local floral resources[40]. This possibly reflects less deviation from

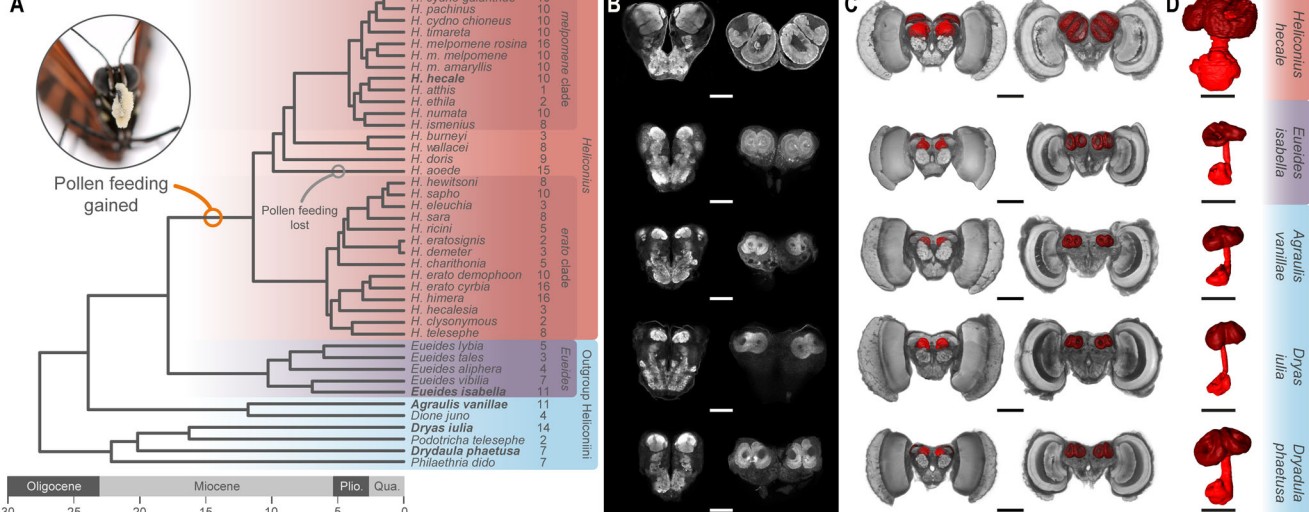

**Fig. 1 | Variation in absolute mushroom body size across the Heliconiini.**
**A** Dated phylogeny showing the Heliconiini species sampled and the major clades, with the number of individuals sampled for each species shown to the right of the species name. **B**, **C** Selected neuroanatomical detail from five species indicated in bold in panel **A**. **B** Confocal scans showing the anterior (left) and posterior (right) of the central brain from one representative individual of each species (scale bars = 250 μm). **C** 3D reconstructions of the whole brain with anterior (left) and posterior (right) views showing the mushroom body (red) (scale bars = 500 μm). **D** Isolated 3D reconstructions of the mushroom body, showing the calyx (dark red), peduncles and lobes (light red) (scale bars = 250 μm). Source data files: *Heliconiini.trees*.

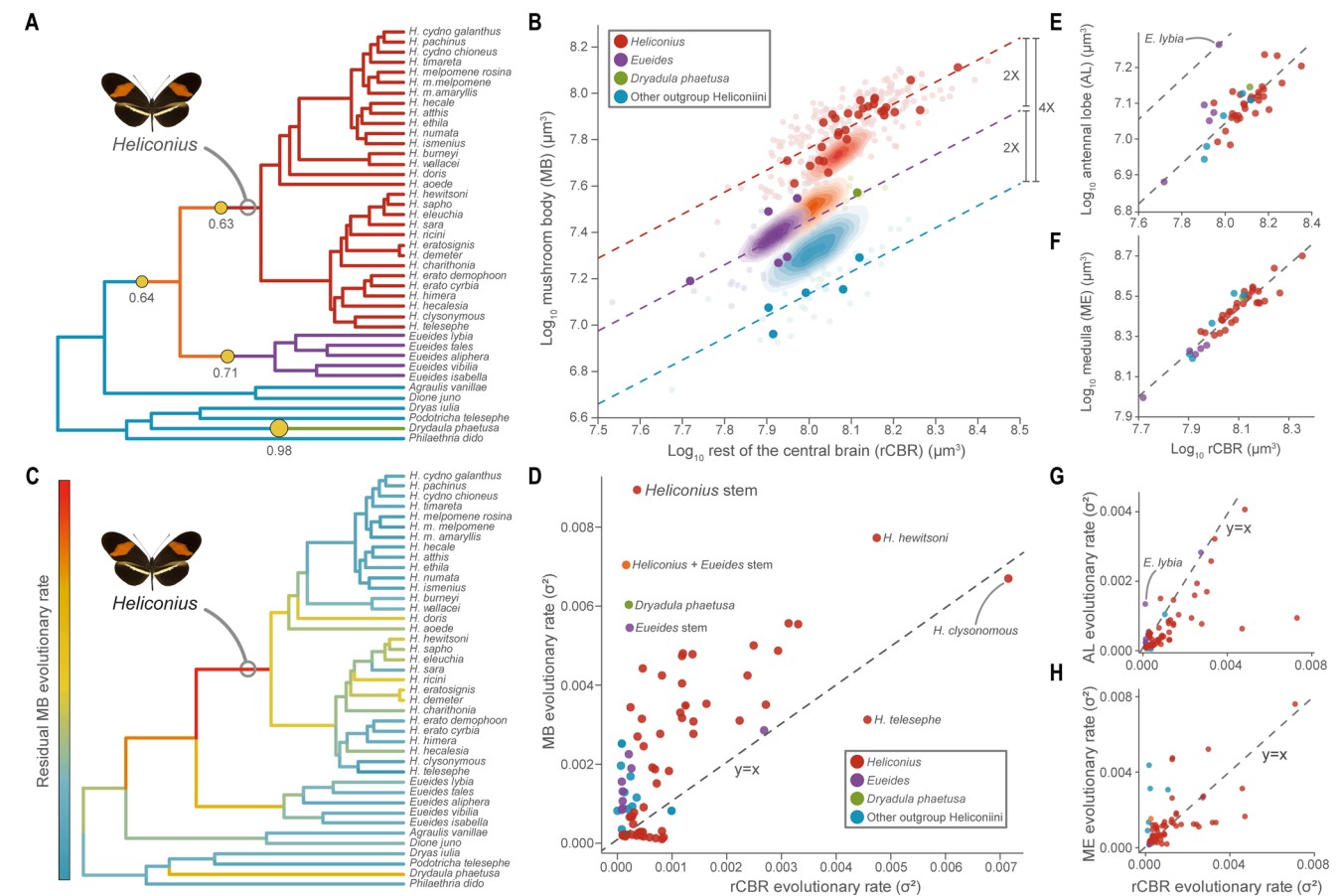

**Fig. 2 | Major shifts in the relative size and evolutionary rate of the mushroom body in *Heliconiini* butterflies. A**, **B** Phylogenetic shifts in the scaling relationship between the volume of the mushroom body (MB) and the rest of the central brain (rCBR) across 41 Heliconiini taxa (posterior probability > 0.5). Relative to outgroup Heliconiini (blue), MB volumes are twice as large in *Eueides* (purple) and *Dryadula phaetusa* (green) and four times as large in *Heliconius* (red). Solid points = species means; faded points = individuals. Estimated ancestral states for each group shown by density maps. **C**, **D** The branch leading to *Heliconius* shows a marked increase in

the evolutionary rate of MB volume, with a slightly less elevated rate along the branch leading to *Heliconius* + *Eueides*. **E**–**H** Shifts in MB size and evolutionary rate are not reflected in either the antennal lobe or the medulla. The butterfly image is from Wikimedia commons, released under CC-BY-SA 4.0. Source data files: **A** Heliconiini.trees and Heliconiini_neuro_species.csv; **B** Heliconiini_neuro_individuals.csv and Heliconiini_neuro_species.csv; **C** Heliconiini.trees and Heliconiini_neuro_species.csv; **D**–**H** Heliconiini_neuro_species.csv.

established trap-lines. In some populations females may also use distinct pollen plants[41].

## Multiple periods of accelerated rates of mushroom body expansion

To further explore the evolutionary history of mushroom body size across the Heliconiini, we used multiple methods to reconstruct ancestral states, evolutionary rates and shifts in allometric scaling. First, we used *bayou*[42] to identify non-allometric shifts in scaling between mushroom body size and central brain size. Such shifts, which are not explained by variation in overall brain size, are generally interpreted as adaptive changes in response to selection pressures on specific brain structures[43]. In these analyses, the best fitting model permitted shifts in elevation specifically (marginal likelihoods: elevation shifts = 54.252; slope and elevation shifts = 37.093; no shifts = 32.556) and identified four shifts with a posterior probability greater than 0.5 (Fig. 2A), representing increases in relative mushroom body size at the internal branches leading to *Heliconius*+*Eueides* (post. prob. = 0.64) and *Heliconius* (post. prob. = 0.63), as well an independent expansion in *Dryadula* (post. prob. = 0.98), and a reduction at the base of *Eueides* (post. prob. 0.71). These shifts result in phylogenetic groupings of species having approximately common allometric scaling of the mushroom body, with convergent allometries between *Eueides*

and *Dryadula*, which are intermediate between *Heliconius* and all other outgroup genera (Fig. 2B). These results are supported by pairwise comparisons among all genera (Fig. S1 and Tables S1 and 2), and by ancestral state reconstructions which again imply that serial shifts in mushroom body size occurred independently of central brain size, and culminated in *Heliconius*, where the estimated ancestral state is within the range of extant *Heliconius* species (Fig. 2B).

We next used evolutionary models that permit branch-specific shifts in rate parameters to estimate points in the Heliconiini phylogeny where either mushroom body or central brain size evolved particularly rapidly. Two alternative methods[44,45] (Fig. 2C, D) identify high rates of evolution for mushroom body size specifically, on multiple internal branches (e.g. Fig. 2C, D). The highlighted branches are the same as in the *bayou* analysis, with the stem *Heliconius* branch standing out as having the highest rate of evolution in mushroom body size relative to central brain size (Fig. 2D). These results provide additional confirmation that expansions in the relative size of the mushroom bodies are a response to targeted selection on these regions, rather than reductions in the size of the rest of the central brain. Sensory brain regions, conversely, do not show similar non-allometric shifts or co-incident periods of accelerated rates of evolution (Fig. 2E–H). This again demonstrates that increases in mushroom body size are not primarily caused by changes in the sensory periphery. However,

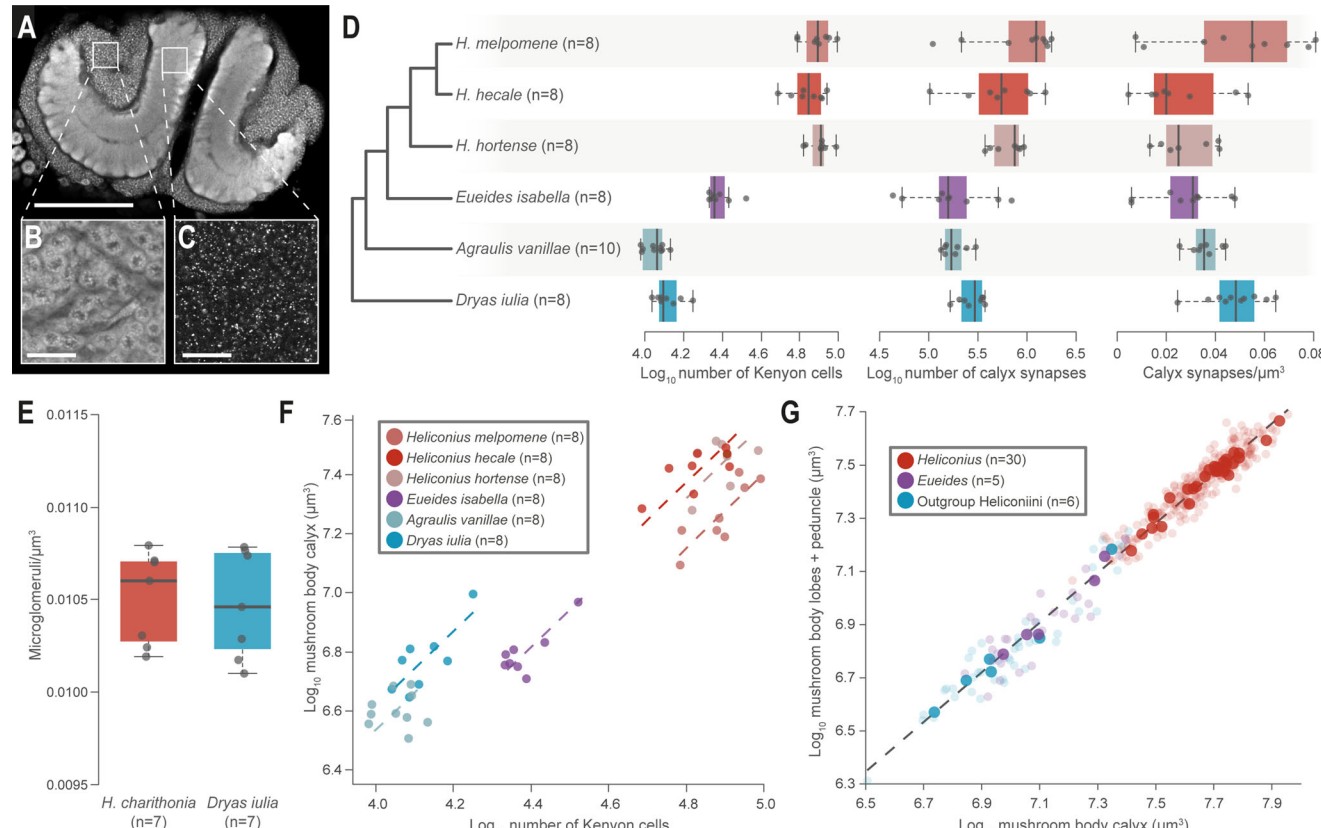

**Fig. 3 | Increased Kenyon cells and synapses numbers in *Heliconius*, but conserved cellular scaling. A** *Heliconius* mushroom body (scale bar = 200 μm), showing (**B**) the intrinsic neurons, the Kenyon cells (KCs) (scale bar = 50 μm) and (**C**) synapses (scale bar = 50 μm). **D** *Heliconius* show increases in the number of Kenyon cells and synapses, but not synapse density. **E** Microglomeruli density does not differ between *H. charithonia* and *Dryas iulia*. In D and E the box encompasses two middle quartiles, with central line showing median. Whiskers extend to the furthest data point within 1.5 times the interquartile range. **F** The relationship

between calyx volume and KC number varies between species, but not overall between *Heliconius* and other Heliconiini. **G** The relationship between the major components of the mushroom body (the calyx and the lobes and peduncle) is conserved across the Heliconiini. Solid points = species means; faded points = individuals. Source data files: **D** KCdata.csv and Synapsedata.csv; **E** phalloidin_means.csv; **F** KCdata.csv; **G** Heliconiini_neuro_individuals.csv and Heliconiini_neuro_species.csv.

whether mushroom body expansion is associated with further circuit changes in downstream neuropils, such as the central complex, which has important roles in spatial navigation in other insects[16], remains to be determined. Finally, to contextualise this variation within a broader sample of butterflies, we used the same approach to reanalyse a phylogenetically broad dataset of 41 species of North American butterflies which includes *Heliconius charithonia* and the non-pollen feeding Heliconiini *Agraulis vanillae*[46]. Both the *bayou* and evolutionary rates analysis highlight the *Heliconius* branch as the sole stand-out lineage, and a remarkably clear outlier in mushroom body evolution across butterflies (Supplementary Note 3 and Figs. S3 and S4).

### Volumetric expansion is closely tied to increases in Kenyon cell number

The volumetric expansions we identify could be due to a combination of (i) increases in the number of Kenyon cells, the intrinsic mushroom body neurons; and/or (ii) increases in the synaptic contacts (branching patterns) made by Kenyon cells, which may result in altered volumes of calyx per Kenyon cell, or increased synapse density or number. To address this, we estimated total Kenyon cell number by staining samples with neural and nuclear markers to measure the volume of the Kenyon cell cluster, which surrounds the posterior calyx (Fig. 3A), and the density of nuclei within it (Fig. 3A–C). We did this for three *Heliconius* and three outgroup Heliconiini species. Our estimates of Kenyon cell number for each hemisphere vary from ~11,000 in *Agraulis vanillae* to ~80,000 in *Heliconius hortense* (Supplementary Data 3).

Total Kenyon cell number varies significantly across species ($\chi^2$ = 1475.3, d.f. = 5, $p$ < 0.0001; Fig. 3D and Supplementary Data 3), and is significantly higher in *Heliconius* ($\chi^2$ = 44.83, d.f. = 1, $p$ < 0.0001). The density of Kenyon cell bodies also varies across species ($\chi^2$ = 34.775, d.f. = 5, $p$ < 0.0001), however, overall it is does not differ consistently between *Heliconius* and other Heliconiini ($\chi^2$ = 0.3617, d.f. = 1, $p$ = 0.548). We also independently verified these estimates by counting cross sectioned Kenyon cell axons running through the mushroom body peduncle, imaged using electron microscopy (Supplementary Note 4 and Fig. S5), in three species ($n$ = 2/species). This produced estimates of ~13,000 Kenyon cells in *Dryas iulia*, ~52,000 in *Heliconius erato* and ~78,000 in *Heliconius melpomene*, which are all within the range of estimates produced using immunostaining.

The scaling relationship between calyx volume and total Kenyon cell number is significantly different across species ($\chi^2$ = 90.749, d.f. = 5, $p$ < 0.0001, Tables S3 and S4), and results in elevation shifts ($W$ = 90.06, d.f. = 5, $p$ < 0.0001). Post-hoc pairwise analysis indicates that this is primarily due to a shift between *Heliconius+Eueides* and other genera, which results in a smaller volume of calyx per Kenyon cell in these two genera (Fig. 3F and Table S5). We also quantified the density of synapses within the calyx to indirectly test for evidence of altered connectivity (Supplementary Data 4). Although we do find an effect of species on synapse density ($\chi^2$ = 17.846, d.f. = 5, $p$ = 0.003), there is no evidence that *Heliconius* consistently differ from other genera ($\chi^2$ = 0.077, d.f. = 1, $p$ = 0.782). However, due to the increased cell number and calyx volume, they do have significantly higher

estimates of total synapse number ($\chi^2$ = 32.664, d.f. = 1, $p$ < 0.0001). We supplemented these data with estimates of the density of microglomeruli (a synaptic complex surrounding a terminal dendrite of a sensory projection neuron[47]; Fig. S6) in *Heliconius charithonia* and *Dryas iulia*. These data also provide no evidence for a shift in density between *Heliconius* and *Dryas* ($\chi^2$ = 0.055, d.f. = 1, $p$ = 0.815; Fig. 3E). Finally, we explored whether the scaling relationship between the calyx, where Kenyon cells synapse with incoming projection neurons, and the lobes, formed by the main axon terminals of Kenyon cells and the dendrites of mushroom body output neurons, is altered in species with expanded mushroom bodies. We found a remarkably consistent scaling relationship across all Heliconiini, with *Heliconius* conforming to the same linear relationship as other genera ($p_{MCMC}$ = 0.180; Fig. 3G).

Together, we propose that these results are consistent with mushroom body expansion being predominantly the result of increased Kenyon cell production and a replication, rather than innovation, of neural circuitry. Current models of insect navigation emphasise the use of egocentric views of visual scenes as a basis for learning spatial information[48]. Theoretical modelling of mushroom body circuitry also suggest that, under conserved levels of average Kenyon cell activity, the capacity to store visual patterns increases logarithmically with Kenyon cell number[33]. Our data therefore indicate that, compared to the ancestral Heliconiini, *Heliconius* likely have expanded mushroom body circuitry to store many more engrams, a unit of cognitive information[49], such as a visual scene, that is imprinted in neural systems. This would provide a substrate for improved navigational performance, across larger spatial scales, through memorisation of large numbers of visual scenes. This ability is likely required for establishing and maintaining long trap-line foraging routes[34].

## Mushroom body expansion is associated with increased visual input

While our data currently imply replication of conserved internal circuitry, the possible link between mushroom body expansion and visually-orientated spatial memory would predict a degree of visual specialisation of mushroom body function. To test this, we first confirmed that visual brain regions send projections to the mushroom body by injecting fluorescent retrograde tracers into the calyx of *H. melpomene*. This revealed two major incoming tracts of projection neurons from the antennal lobe and the optic lobe (Fig. S7). In the optic lobe, staining was diffuse across multiple structures, but more concentrated in the ventral lobula, which we suggest may act as a relay centre to the central brain (Supplementary Note 5 and Fig. S7). Notably, this structure is also highly variable in size (31.6-fold variation), but is not specifically expanded in *Heliconius* ($p_{MCMC}$ = 0.482; Fig. S8). We next differentially traced sensory projections from visual and olfactory neuropils to record the location and volume of calyx receiving input from each sensory modality, in eight species, including four *Heliconius* and four outgroup Heliconiini. In all species, the calyx is topographically segregated by sensory input (Fig. 4C–H), i.e. visual and olfactory projection neurons terminate in largely non-overlapping areas of the calyx, as has been observed in some other butterflies[50]. This enabled segmentation of discrete areas of calyx receiving visual or olfactory input (Supplementary Data 5). The volume of both the olfactory ($\chi^2$ = 10.396, d.f. = 1, $p$ = 0.0012. Fig. 4I) and visual ($\chi^2$ = 33.8, d.f. = 1, $p$ < 0.0001, Fig. 4I) calyx are expanded in *Heliconius*. However, variation in the visual calyx volume (Cohen's $d$ = −10.1) is considerably larger than for the olfactory calyx (Cohen's $d$ = −3.24, Fig. 4I). The scaling relationship between these two regions is also significantly different across species ($\chi^2$ = 700.74, d.f. = 7, $p$ < 0.0001), with *bayou* identifying a major shift between *Heliconius* and other genera (post prob = 0.7; Fig. 4I). This results in *Heliconius* having visual calyx volumes ~2-fold larger than would be predicted by olfactory calyx

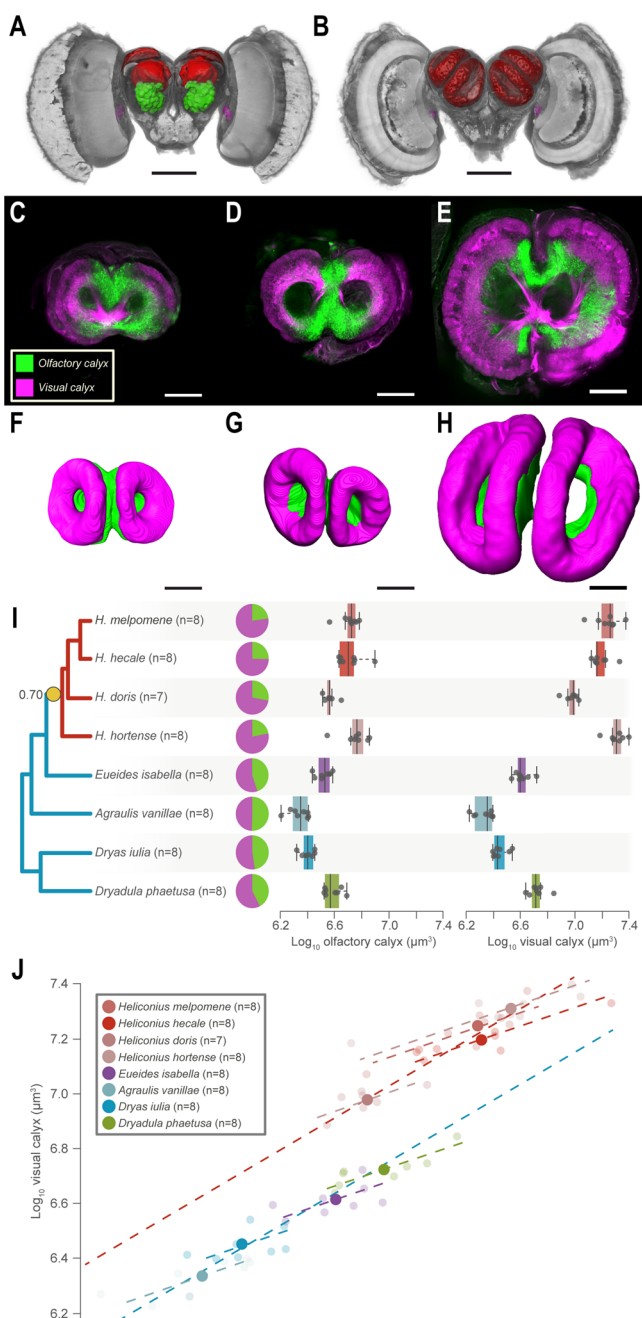

**Fig. 4 | Mushroom body expansion in *Heliconius* driven primarily by increased visual input. A** Anterior and **B** posterior 3D-reconstructions of *H. hecale* brain showing injection sites for the tracing of sensory projections to the MB (red neuropil). Olfactory projections neurons were traced from the antennal lobe (green neuropil) while visual projection were traced from injections around the ventral lobula (magenta). Scale bars in **A** and **B** = 500 µm. **C** Optical section of the mushroom bodies acquired with a confocal microscope in *Dryas iulia*, **D** *Eueides isabella* and **E** *H. hecale*. Inputs from olfactory (green) and visual (magenta) sensory neuropils terminate in segregated regions of the calyx. **F**–**H** 3D volumetric reconstructions of the visual and olfactory regions of the calyx in (**F**) *Dryas iulia*, (**G**) *Eueides isabella* and (**H**) *H. hecale*. Scale bars in **C**–**H** = 100 µm. **I** *Heliconius* exhibit a volumetric increase in the olfactory region of the mushroom body, but a greater increase in the visual region. The box encompasses two middle quartiles, with central line showing median. Whiskers extend to the furthest data point within 1.5 times the interquartile range. **J** Controlling for olfactory calyx volume, *Heliconius* exhibit an upshift in size of the visual calyx. Solid points = species means; faded points = individuals. Sample sizes are indicated in the figure 'n' = number of individuals/species. Source data files: **I**, **J**: Tracingdata.csv.

volume compared to other Heliconiini genera. Among the outgroup genera we also observe post-hoc shifts in scaling that suggest increased visual input may contribute to smaller scale variation in mushroom body expansion (Supplementary Note 5 and Tables S6 and 7). However, the shift in *Heliconius* is, again, more extreme than this pattern would predict (Fig. 4J). This demonstrates that a major change in the degree of visual processing performed by the mushroom bodies coincided with the origin of pollen feeding. Notably, our data suggest this marked expansion of visual processing is not repeated in other periods of mushroom body expansion, such as in *Dryadula* or the stem *Heliconius+Euiedes* branch.

## Ecological explanations for mushroom body expansion

Increased visual processing is consistent with a plausible link between mushroom body expansion and visually-orientated spatial memory. However, our comparative analysis reveals multiple periods of mushroom body expansion across the phylogeny. Ancestral state reconstructions of the presence/absence of pollen feeding currently imply a likely single origin for this trait at the base of *Heliconius*, with a secondary loss in *H. aoede* (Supplementary Note 7 and Fig. S9). While the presence of pollen feeding alone (DIC = −840.512) and a general genus effect (DIC = −840.892) have equal power in explaining variation in mushroom body size, the gain of pollen feeding coincides with the highest rate of mushroom body evolution (Figs. 2, S3 and S4) and the dominant shift in visual innervation to the calyx (Fig. 4J) strongly implicating this innovation as a causative agent in mushroom body expansion. We do not identify a secondary reduction in mushroom body size in *H. aoede*, the sole *Heliconius* lineage to have putatively lost pollen feeding (Fig. S1; pMCMC = 0.543). However, the foraging behaviours of this lineage are unknown, its phylogenetic position remains contentious[9], and the clade also has unique host plant associations, being specialised on *Dilke*a, a genus of Passifloraceae with tree-like growth forms, a shift which may have altered their wider cognitive ecology. Regardless, this suggests that large mushroom bodies were not counter selected with the probable loss of pollen feeding in this lineage. We therefore explored whether additional traits which could plausibly be linked to mushroom body function are also associated with variation in mushroom body size across the tribe (Supplementary Data 6). First, Heliconiini share larval host plants from the Passifloraceae, and levels of host plant generalism could explain some variation in mushroom body size and plasticity[51,52]. However, using a dataset of host plant records[53] we found no association between the number of host plants used and mushroom body size (pMCMC=0.492). Second, some Heliconiini form aggregated roosts at night, and it has been argued that this could facilitate social transfer of information on resource location[54] and exert distinct selection pressures on the processing of conspecific cues in social species. However, we again found that the degree of social roosting had no power to explain variation in mushroom body size (pMCMC = 0.857). One additional explanation for the independent increases in mushroom bodies in non-trap lining species, such as *Eueides* and *Dryadula*, is that they possess an ability to form visual memories as part of their wider behavioural repertoire. For example, *Heliconius* have strong site fidelity and homing ability, a likely pre-requisite for trap-lining, but also a trait shared with at least some territorial species of *Eueides*[55]. True site fidelity likely requires a degree of spatial memory[56,57], and mushroom body expansion could therefore reflect an increase in the capacity to store more of these memories. We therefore suggest variation in this function may explain independent shifts in mushroom body size, and provide the foundation for the extreme expansion observed in *Heliconius*. Unfortunately, current ecological data on the movement ecology of non-*Helconius* Heliconiini is limited, prohibiting formal tests of this hypothesis.

## Evidence for increased precision in visual discrimination in Heliconius

The increase in visual projection to the mushroom body calyx also provides a clear prediction that increases in mushroom body size should be associated with enhanced performance in some visual learning and memory contexts[33]. *Heliconius* are capable of associative learning (A + , B-) between a reward and either colour[58–60], shape[61], or odour cues[60], as well as contextual cues such as time of day[59]. Natural foraging, however, likely involves encounters with complex combinations of cues. To begin to explore visual discrimination in foraging Heliconiini, we therefore focused on two "non-elemental" learning tasks[62]; positive pattern learning (A-, B-, AB + ) and biconditional discrimination (AB + , CD + , AC-, BD-). Using artificial, coloured feeders, we trained individuals of *Dryas iulia* and *Heliconius erato*, as representatives of species with small and large mushroom bodies respectively, to solve these tasks in insectary conditions (Supplementary Data 7). We found that both *D. iulia* ($Z$ ratio = −9.182, $p < 0.0001$) and *H. erato* ($Z$ ratio = −16.396, $p < 0.0001$) can solve positive patterning tasks. However, a significant interaction between species and trial indicates that *H. erato* are more accurate in their post-training performance ($\chi^2 = 66.533$, d.f. = 1, $p < 0.0001$; Fig. 5A). A smaller sample ($n = 13$) of *H. melpomene* suggest similar performance levels within *Heliconius* (Supplementary Note 8, Fig. S10 and Table S8). In the more challenging biconditional discrimination task this interaction is also found ($\chi^2 = 20.727$, d.f. = 1, $p < 0.0001$) with *H. erato* ($Z$ ratio = −5.465, $p < 0.001$), but not *D. iulia* ($Z$ ratio = −1.241, $p = 0.601$), being able to learn these complex cue combinations (Fig. 5B and Supplementary Data 8). Empirical studies in other insects suggest that mushroom bodies are essential for the ambiguous discrimination of configural cues[35,36]. Computational models of mushroom body circuitry also suggest solutions for these more complex tasks can emerge from circuits supporting simple associative learning (A+, B−), and that synaptic plasticity between projection neurons and Kenyon cells may be critical in mediating this process[63]. Our results suggest that increases in the amount of visual projection neurons to the mushroom body calyx in *Heliconius*, and/or the associated greater number of Kenyon cells and synapses, may also affect this computation. We propose that these results may reflect increased precision in comparisons of visual scenes used during navigation, through sparse coding across larger number of neurons. Thus, complex panoramic image of landscapes, which are variable and specific arrangements of shapes and colours, would be integrated across more units, resulting in less overlapping coding of visual configurations. Currently, however, we cannot formally link enhanced performance in our configural discrimination experiments to trap-line foraging, and it is possible this ability could be co-incident to traits under direct selection for trap-lining. For example, the restricted range of pollen plants used by *Heliconius*[64] could be supported by selection for increased fine discrimination of floral cues.

## Evidence for increased long-term memory retention in Heliconius

The extended longevity of *Heliconius* butterflies[11], combined with the stability of their home range[65] and preferred floral resources[54], means that learnt foraging routes can be utilised for several months[14]. *Heliconius* foraging routes may therefore persist longer than the natural lifespan of other Heliconiini. Mushroom bodies are central to long-term olfactory memories in insects[66,67], and while their role in long-term visual memory is less explored, it is likely conserved across modalities. We therefore hypothesised that *Heliconius* may possess more stable long term visual memories, as part of the suite of behavioural traits that accompanied the evolution of pollen feeding, increased longevity and mushroom body expansion. To test this hypothesis, we again trained *D. iulia* and *H. erato* on artificial feeders, this time in a simple two-colour preference assay (purple/yellow)

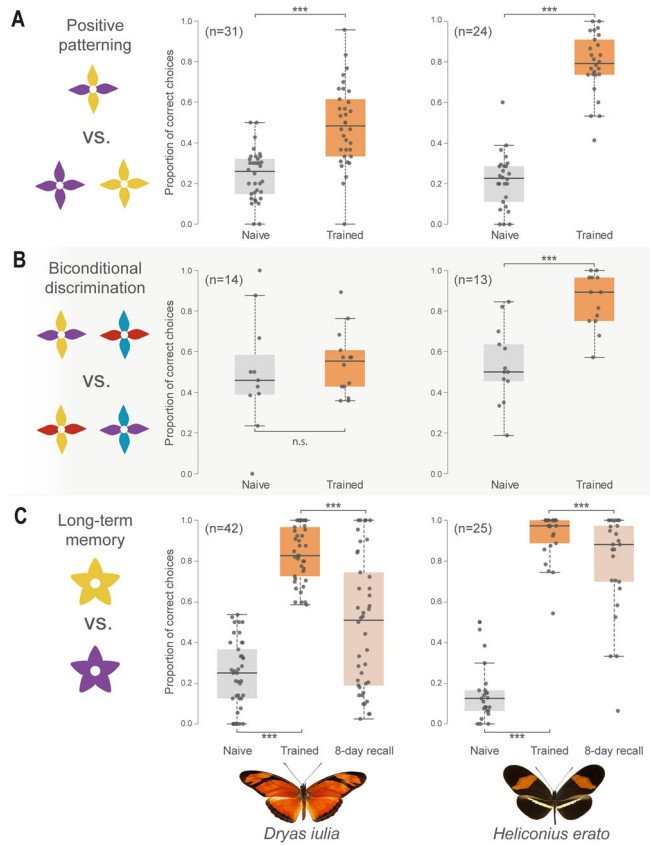

**Fig. 5 | Enhanced non-elemental visual learning and long-term memory in *H. erato* relative to *Dryas iulia*. A** Both species solve a visual positive patterning task, but *H. erato* is significantly more accurate. **B** *H. erato* can solve the more difficult and strictly non-elemental biconditional discrimination task, whereas *Dryas iulia* cannot. **C** *H. erato* has superior memory of a learned visual cue after 8 days compared with *Dryas iulia*. The box encompasses two middle quartiles, with central line showing median. Whiskers extend to the furthest data point within 1.5 times the interquartile range. Data were analysed with generalised linear models treating species and training as fixed effects along with an individual-level random effect. Posthoc comparisons were made by deriving the estimated marginal means and using a two-sided z-test, correcting for multiple comparisons using Tukey's test. *$p < 0.05$; **$p < 0.01$; ***$p < 0.001$. Source data files: **A**: pospatdata.csv; **B**: bicondiscdata.csv; **C**: LTMdata.xslx.

(Supplementary Data 9). After demonstrating that both *H. erato* (Z ratio = −12.136, $p < 0.001$) and *Dryas iulia* (Z ratio = −11.321, $p < 0.001$) learn to associate a reward with a non-preferred colour cue (Fig. 5C), we fed the butterflies with neutral (white) feeders for eight days. In subsequent preference trials where individuals were again presented both purple and yellow feeders, *D. iulia* performed significantly worse than *H. erato* in recalling the positively reinforced colour (Z ratio = −4.829, $p < 0.0001$). *H. erato* maintains a high accuracy towards the reinforced cue (Z ratio = 4.635, $p < 0.0001$), while *D. iulia*'s preference is no longer significantly different to random (Z ratio = −0.160, $p = 0.873$; Fig. 5C). We suggest that these results are indicative of longer retention of visual memories in *Heliconius*, which would support the stability and long-term exploitation of learnt foraging routes.

**A phylogenetic view of mushroom body expansion**
By combining extensive, phylogenetically dense sampling of wild individuals, comparative analyses of the rate of neural evolution, and focused behavioural experiments, our data demonstrate considerable variation in a prominent brain structure within a closely related and - except for the suite of traits associated with the dietary

innovation of pollen feeding – ecologically comparable tribe of Neotropical butterflies. Notably, the expansions we observe in this tribe, which diverged across the last ~25 mya, are an order or magnitude more recent than other major mushroom body expansion events[68], meaning comparative studies suffer from fewer confounding effects. Taking this approach, we provide evidence of mushroom body expansion associated with more extensive visual input to the calyx, and an eight-fold increase in Kenyon cell number. The presence of multiple periods of mushroom body expansion suggests the ecological factors which favour increased investment in mushroom bodies are multifaceted. However, we provide evidence that the most substantial shifts in size and sensory modality co-occur with the origin of pollen feeding and trap-line foraging. These neuroanatomical shifts were accompanied by enhancements in long-term visual memory, and a capacity to discriminate more complex visual patterns. We hypothesise that these behavioural differences reflect either a direct response to selection on foraging behaviour, correlated traits such as increased longevity or floral specificity, or an indirect consequence of increased Kenyon cell number. Combined, our results emphasise the intimate relationship between the structure and function of a species' nervous system and natural ecology. Our data also highlight Heliconiini as a tractable system for comparative, detail-rich analyses aimed at understanding the adaptive and mechanistic basis of neural and behavioural evolution.

## Methods
This research complies with all relevant ethical regulations and collection permits as provided by the relevant authorities in Costa Rica, Panama, French Guiana, Ecuador and Peru. Relevant permit numbers are provided in the Supplementary information, along with more technically detailed methodologies.

### Animals
To build an extensive dataset of neuroanatomical traits across the Heliconiini tribe, we caught 318 wild butterflies from 41 species from sites in Costa Rica, Panama, French Guiana, Ecuador and Peru (Supplementary Method 1). For measurements of the cellular composition and sensory innervation of the calyx, we obtained pupae from 8 species representing key lineages, from commercial suppliers. These butterflies were reared in controlled temperature room at 28 °C with 80% humidity level and a 12-h light cycle.

### Neuroanatomical staining
Brain preservation and subsequent neuropil staining was performed following an established protocol[69] (Supplementary Method 2). Brains were dissected out under isotonic buffer-saline (HBS) and fixed for 16–20 h in zinc-formalin solution (ZnFA). Prior to long-term storage at −20 °C in methanol, brains were incubated in a solution of 80% methanol and 20% DMSO for 2 hours. When ready to process, brains were rehydrated in a series of decreasing methanol concentrations in Tris buffer. For measurement of neuropil volumes and presynaptic boutons density, brains were immuno-labelled with a combination of anti-synapsin (primary antibody, mouse anti-SYNORF1) and Cy2 conjugated secondary antibody (cy2 goat anti mouse IgG). After a dehydration series, brains were clarified, stored and imaged in methyl salicylate (Supplementary Method 3).

Neuronal populations were traced to analyse variation in the sensory domains within the calyx of 63 individuals from 8 representative species (Supplementary Method 4). Butterflies were first cold anaesthetised and immobilised in custom-made holders. The brain was then exposed under Ringer solution, and dextran-conjugated dyes (fluoro-ruby and Alexa 647) were injected into the primary sensory neuropils, the antennal lobe and the optic lobes. After being kept in the dark overnight, the brain was dissected, fixed and stained with anti-synapsin as described above.

Kenyon cell number was assessed in 50 individuals from 6 representative species, reared under controlled conditions (Supplementary Method 5 and 6). After rehydration from storage, the brains were embedded in agarose and sliced into 80 μm frontal sections. The tissues were immuno-labelled with a combination of anti-peroxidase antibody (HRP: Rabbit anti-horseradish peroxidase) and Cy3-conjugated secondary antibody (Cy3 goat anti-rabbit IgG), while cell nuclei were stained with DAPI. Data on synapse number was estimated from the same individuals, with the addition of 7 individuals of *H. charithonia* and *D. iulia* stained with anti-SYNORF1 (as above) and Alexa 488 phalloidin (A12379, Thermo Fisher Scientific) to reveal both post and presynaptic (microglomeruli) structures (Supplementary Method 5 and 7).

## Image acquisition and processing

Whole mount brains were scanned in methyl salicylate using a confocal laser-scanning microscopes (Leica SP5 or SP8) with a ×10 dry objective (0.4 NA), while the cellular composition of the calyx was examined with a ×63 glycerol immersion objective (1.3 NA; Supplementary Method 8). According to the immersion medium, image stacks were rescaled along the z-dimension to correct for axial shift of the emitted light. 3D models of neuropils were generated using Amira 5.4.3. Kenyon cells were counted in $25 \times 25 \times 15$ μm boxes and the plugin 3D Object Counter from ImageJ was used to automatically count synaptic boutons in $50 \times 50 \times 15$ μm boxes (Supplementary Method 9). To validate Kenyon cell counts we also imaged cross sections of axons running through the peduncle using Electron Microscopy (Supplementary Method 10).

## Behavioural experiments

Behavioural experiments used captive-reared butterflies and were conducted in 2x2x3 m mesh cages. Freshly eclosed butterflies were first introduced to a pre-training cage containing only white feeders filled with sugar-protein solution for two full days to accustom individuals to the use of artificial feeders.

For the positive patterning assays, initial preference was tested by introducing butterflies to a cage with empty feeders coloured in yellow, purple and yellow + purple (4 feeders each) and counting feeding attempts for 4 h. During the training phase, the two-coloured feeders (yellow + purple) were filled with a sugar-protein reward while single-coloured yellow and purple feeders were filled with an aversive solution of quinine. Butterflies could freely sample the feeders for eight days prior being re-tested for feeding preference to assess learning performance.

For the biconditional learning assays, after acclimation on white feeders, the initial preference was tested, as previously, on empty artificial feeders of four different colour combinations: red + blue; purple + yellow; red + yellow; and purple + blue. During the training phase, feeders were filled with a sugar-protein reward or quinine following one of two possible combinations (purple + yellow and blue + red vs. yellow + red and blue + purple). Accordingly, the task could not be solved by learning a single colour, but required learning a specific colour combination. As for the positive patterning assay, the butterflies could freely sample the feeders for eight days before re-testing.

For the long-term memory (LTM) assay, initial preferences were tested on empty feeders of either purple (12) or yellow (12) colour. Butterflies were then trained for four days to associate a food reward with their non-favoured colour, based on the performances in their initial preference test. Next, the butterflies were re-tested to verify the acquisition of the colour-food association. Individuals were then placed in a "pre-training cage" containing only white feeders filled with a sugar-protein solution for eight days. To test for memory retention of the colour association acquired during training, butterflies were subjected to a third preference test.

## Statistical analysis

All statistical analyses were performed in R v 4.1.2. Ecological data were collated from the literature (Supplementary method 11), and phylogenetic comparative analyses were conducted using a new phylogenetic tree of the Heliconiini generated from newly assembled genomes[70] (Supplementary method 12). To assess how the volumes of neuropils vary across the Heliconiini, we ran a series of phylogenetic generalised linear models (GLMMs) with gaussian distributions using the R package *MCMCglmm v 2.32*[71], using the rest of the central brain (rCBR) as an allometric control. Ecological explanations for mushroom body expansion were tested by taking the best fitting of these models and then including ecological factors – the degree of social roosting, host plant number and pollen feeding. The R package *bayou* v 2.0[72] was used to identify regions of the Heliconiini tree showing evidence of a shift in the scaling relationships between specific neuropils and rCBR. rCBR is used as an allometric control as it excludes all neuropils of interest (MB, AL, MED, vLOB) and therefore allows comparison of these results while excluding their influence on one another. Non-allometric "grade-shifts" are widely viewed as signatures of adaptive change in brain composition, reflecting a response to selection pressures acting on specific brain components[43]. The *bayou* analyses were supplemented by an analysis of allometric shifts between mushroom body and rCBR volumes using the R package *smatr* v 3.4-8[73]. We subsequently used two methods to test for shifts in the evolutionary rate of change in mushroom body size within the Heliconiini. First, we used *BayesTraits* v3 to compare two independent contrasts MCMC model of evolution, one allowing for a rate scaling parameter to vary across branches, and one with a fixed average rate[74]. Second, we used a recently-published method that uses Brownian motion to model variations in evolutionary rates included in the R package *phytools* v 0.7-90[45]. These analyses allow us to confirm that the grade-shifts identified by *bayou* are the result of variation in the rate of MB evolution, rather than rCBR evolution, supporting the inference of adaptive MB evolution.

Ancestral states for mushroom body size and rCBR at key internal nodes in the Heliconiini tree were also estimated using two methods. First, using *BayesTraits* we estimated ancestral values using a non-directional model and the scaled trees generated from the previous varied rates analyses. Second, we used the *fastAnc* function in *phytools* to estimate the maximum likelihood ancestral states for mushroom body and rCBR volumes at each node. The presence of pollen feeding at internal nodes in the Heliconiini tree was estimated using three different methods: (1) MCMC stochastic character mapping in *phytools*[75] (2) maximum likelihood using the *ace* function in the R package *ape* v 5.5[76] and (3) maximum parsimony using the *asr_max_parsimony* function in the R package *castor* v1.7.0[77].

Variation in the number of Kenyon cells, calyx synapses, microglomeruli, and in the volumes of the visual and olfactory calyces were explored using a series of generalised liner models (GLMs) and GLMMs, using the *glm* and *glmer* function from the R package *lme4* v 1.1-30[78], respectively. For interspecific differences, species was treated as a fixed effect, and for differences between *Heliconius* and outgroup Heliconiini, species was treated as a random effect and *Heliconius* membership as a fixed effect. Pairwise differences were assessed by calculating the estimated marginal means using the function *emmeans* in the R package *emmens* v 1.7.0 and correcting for multiple comparisons using the Tukey test[79]. Analysis of variation in the scaling relationship between the visual and olfactory calyces was additionally tested for using the function *sma* from the R package *smatr*[73] and the R package *bayou*[72].

Recall performance in the behavioural assays was analysed with GLMMs using the *glmer* function from the R package *lme4*[78]. These models included species and training as fixed effects (in addition to their interaction), with an individual-level random effect. All post hoc comparisons were made by obtaining the estimated marginal means

using the R package *emmeans* and were corrected for selected multiple comparisons using the Tukey test[79].

## Reporting summary

Further information on research design is available in the Nature Portfolio Reporting Summary linked to this article.

## Data availability

All data generated in this study are available in the Supplementary Information and have been deposited in the DataDryad database available at https://doi.org/10.5061/dryad.f1vhhmh28. Source data are provided with this paper.

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

## Acknowledgements

We are very grateful to the environmental ministries of Costa Rica, Panama, French Guiana, Ecuador and Peru for permission to collect and export samples. We thank the Organization for Tropical Studies, Le Leona Eco Lodge, the Smithsonian Tropical Research Institute, the Estación Científica Yasuní, PUCE, F. Ramirez Castro, Neil Rosser, Ronald Mori Pezo the Dasmahapatra group, and the broader *Heliconius* research community for support in the field and for discussions, and to Swidbert Ott for advice and encouragement early on in this project. We are grateful to the Wolfson Bioimaging Centre, University of Bristol, the University College London Confocal Imaging facility, and Matt Wayland and the Dept. of Zoology Imaging Facility, University of Cambridge, for imaging assistance. This work was supported by a Royal Commission for the Great Exhibition Research Fellowship (S.H.M.), a Leverhulme Trust Early Career Fellowship (S.H.M.), a Short-term STRI Fellowship (S.H.M.), a Royal Society Research Grant (S.H.M.), a Newton Trust INT Research Grant (S.H.M.), a NERC Independent Research Fellowship NE/N014936/1 (S.H.M.), an ERC Starter Grant 758508 (S.H.M.) and a PhD Studentship from Trinity College, Cambridge (F.J.Y.).

## Author contributions

Conceptualisation: S.H.M.; Methodology: S.H.M., A.C., F.J.Y., C.N. and F.C.; Investigation: A.C., F.J.Y., D.A., S.M., L.M.F., L.H., M.M., C.N. and F.C.; Visualisation: F.J.Y. and A.C.; Funding acquisition: S.H.M; WOM Project administration: S.H.M., WOM supervision: S.H.M., WOM writing – original draft: S.H.M., A.C. and F.J.Y.; Writing – review and editing: S.H.M., A.C., F.J.Y., W.O.M., F.C., C.N., D.A., S.M., L.M.F., L.H. and M.M.

## Competing interests

The authors declare no competing interests.
