## [Peer Review File · Nature Communications]

Rapid expansion and visual specialisation of learning and memory centers in the brains of Heliconiini butterfliesREVIEWER COMMENTS

Reviewer #1 (Remarks to the Author):

Dear editors, dear authors

This work presents strong evidence for a link between the evolution of pollen feeding (at the base of the *Heliconius* genus) and mushroom bodies (MB) expansion. Further evidence shows convincingly that this expansion in MB size is due to a higher number of Kenyon Cell (rather than a change in MB structure), and correlates with an increased amount of visual projection to the MB. What's more, the authors demonstrate that this brain evolution is not a response to a change in the number of host plants, or the degree of social roosting, but is tightly link to the evolution pollen feeding which is known to involve trap-lining behaviours and thus the need for long route visual learning. Literature in ants and bees as shown indeed that long-route learning requires a large amount of KCs (receiving visual input), providing a very pleasant rationale to the story.

The amount of data presented, as well as the work necessary to produce them is extremely consequent. The idea that MB evolved to support visual navigation was already in the literature for quite some time (see Webb and Wystrach 2016 for a review) mostly due to work in central place foraging hymenopteran; but the evidence was rather sparse. This work therefore provides the strongest support for this idea, and what's more, demonstrate it in an entirely different group of insects, and through a rapid radiation that is much more recent and with a clearly identified behavioural need (route following for pollen feeding in butterflies). I have no doubt that this study will stand as a hallmark of our understanding of the evolution of brains in insects and I therefore strongly recommend it for publication in *Nature communication*.

This said, I have a single, major concern that could be easily alleviated.

Major point

There seem to be an apparent confusion in the ecological explanation provided. A confusion between, on the one hand the need for a large amount of Kenyon cells (and hence large MB) to increase memory space for visually learning long route; and on the other hand the general term of 'cognitive performance' and the ability to solve experimental tasks such as the one you tested here (ambiguous color-flower discrimination testing the notion of configural learning).

It is true that both visual route following and experimental paradigm for configural learning, require the MB; however, it is really not clear whether the latter is required for the former (as implied here), or whether they have anything to do with each other (apart from requiring the MB). On the contrary, our current understanding of route following is that it requires the egocentric learning of wide visual field input (the whole scene) without extraction of its individual components. Large KC number are required to store more visual scene, and thus longer route. This is something quite different that the ability for fine and ambiguously coloured flower recognition (which may indeed be helped by larger KCs enabling a sparser coding, and thus affording more discrimination).

I see here therefore two different functions, one is about the evolution of memory space (for long route learning), and one is about the memorisation of fine visual discrimination (required to solve the paradigm presented here). I would thus recommend to present these as alternative hypothesis, rather than attempting to mingle them into a single one. One evolutionary scenario (which seem to be the one backed up in the conclusion) is that 'high cognitive performance' is required for traplining (we don't have evidence for that); the other is that traplining require learning long routes and thus a high number of visual scene, which requires a large memory space (hence lots of KCs), and that large amount of KCs provide, as a by-product, a better ability to solve experimental task requiring to discriminate and memorise ambiguous visual pattern.

All details and references regarding these points are provided in the minor comments below.

Minor points

-L51. This dietary innovation is accompanied by the evolution of trap-line foraging, where individuals learn foraging routes between resources with high spatial and temporal fidelity^{5,14,15}. Among insects, this foraging strategy is also found among species of bee^{16,17}, and is thought to rely on visual memories and landmark cues¹⁸.

The first sentence is absolutely right, but the second is a bit confusing. It is not strictly true that trap-lining requires the learning of visual cues. There are models showing that this ability requires before all the ability to form celestial vector memory (involving rather the central complex, see LeMoel et al., 2019). However, it is indirectly true: Trap-lining requires the memorisation of long routes and multiple locations, and this, in turn (whether traplining or not) is known to require the memorisation of a large amount of visual scenes, as shown in ants (who don't trapline) and bees (who do trapline).

I think this subtlety is important to convey, as the argument that large amount of KCs is needed for learning visual scenes for navigation comes before all from studies in ants.

Le Moël, F., Stone, T., Lihoreau, M., Wystrach, A., & Webb, B. (2019). The central complex as a potential substrate for vector based navigation. *Frontiers in psychology*, 10, 690.

-L 60 However, knowledge of how the adaptative evolution of sensory integration and learning centres supports cognitive enhancement is limited.

Introduction jump between the rationale for the need for better 'spatial orientation' skills or 'cognitive enhancement'. It is unclear how this two terms relate in their rationale. What selective pressure is thought to have increase the MB (see major point).

-L58. While not essential for spatial memory in *Drosophila*²², mounting evidence from empirical^{23–25} and theoretical modelling^{26,27} implicates these structures in spatial orientation in other insects.

Ref 27 (Baddeley et al., 2012) is not appropriate, as it is not a mushroom body model. However, Webb and Wystrach 2016 review the evidence for the role of large MB bodies in insect visual navigation.

Webb, B., & Wystrach, A. (2016). Neural mechanisms of insect navigation. *Current Opinion in Insect Science*, 15, 27-39.

-L76 mushroom body volume is associated with variation in both visual and olfactory structures (medulla: pMCMC = 0.002; 77 antennal lobe: pMCMC < 0.001).

I assume the correlation is positive? Would be good to make this clearer.

-L 84. Interestingly,⁸⁴ we also identify a *Heliconius* specific effect of sex (interaction *Heliconius**Sex pMCMC < 0.001),⁸⁵ with females having larger mushroom bodies on average than males (pMCMC < 0.001; Figure S2), which could reflect sex differences in foraging for pollen or host plant resources²⁹.

Interesting indeed. I would however mention here what are these differences in foraging (from ref 29). Is the evidence in the good direction (with females foraging across more locations, or needing longer route) ?

-L91 First, we used BAYOU30 to identify shifts in scaling between mushroom body size and central brain size.

I am not qualified to assess the pertinence of the phylogenetic analysis (such as the use of BAYOU). However, it would be good to add a sentence to justify why the MB size was scaled relative to the central brain size (the latter being used as a control I assume) rather than the medulla, AL or overall brain size. But can this method disentangle between an evolutionary increase in MB size vs. a decrease in central brain size ?

It is concluded L103 'change in mushroom body size independently of central brain size. But, apart from the fact that the MB variation is greater than the central brain variation, it is not yet clear to me how the opposite hypothesis (change of central brain size independently of MB size is rejected based on this approach).

The subsequent phylogenetic analyses (rate based, L105 fwd) however seem to answer my above concern. My question is then: why not presenting the rate analyses first (based on absolute size), and then, as a second step, back up the results with the Bayou analysis (based on relative size) ?

-L 168 To test this, we first confirmed that visual brain regions send projections to the mushroom body by injecting fluorescent retrograde tracers into the calyx.

It seems (Fig S8) that this analysis was conducted with the single species (*H. Melpomene*). I don't think this is a problem, but it should be mentioned in the main text.

-L218. One outstanding explanation is that the cognitive processes required for trap-line foraging are co-opted or refined from those supporting other, related, behaviors

I am not sure I understand this point. I believe this sentence is unclear because the concept behind is unclear. Why not simply suggesting that the need to learn longer routes and hence to form more visual memories of the scenes, be the reason? (see major point)

-L244 Configural learning, particularly the integration of multisensory cues, plays a crucial role in insect navigation^{25,48}, and these results suggest that mushroom body expansion in *Heliconius* may support improvements in this ability.

I am not convinced by the pertinence of this statement. First of all, 'configural learning' as tested here (with bicolour plants) is not a multisensory task. Second, I am not sure the evidence cited support the claim. Ref 25 (Bulehman et al., 2020 in ants) do not focus on configural learning at all; and Ref 48 (Flanigan et al., 2021, in whip spiders) concerns a discrimination task, which has not much to do with navigation (and even less insect navigation or visual navigation).

However, there is direct, albeit independent, evidence in insects that: on the one hand, visual navigation and notably long route following requires (and do correlates with) large amount of KCs (reviewed in Webb and Wystrach 2016); and on the other hand, that cognitive experimental paradigms such as the one you tested here (testing for the concept of configural learning) also requires the MB (Devault et al., 2015; Durrieu et al., 2020). However, the two type of tasks are quite different, and it is not clear how they converge ecologically (apart from the fact that they both require the MB).

More generally, it is not clear how the paradigm you used here with coloured flower-shaped pattern discrimination can relate directly to the ability to follow long routes. However, it demonstrates clearly a superior ability of the Heliconus species tested here to discriminate between ambiguous flowers using colour vision. Now the question is, does this relate to pollen feeding? And if yes, how ?

Devaud, J. M., Papouin, T., Carcaud, J., Sandoz, J. C., Grünewald, B., & Giurfa, M. (2015). Neural substrate for higher-order learning in an insect: mushroom bodies are necessary for configural discriminations. *Proceedings of the National Academy of Sciences*, 112(43), E5854-E5862.

Durrieu, M., Wystrach, A., Arrufat, P., Giurfa, M., & Isabel, G. (2020). Fruit flies can learn non-elemental olfactory discriminations. *Proceedings of the Royal Society B*, 287(1938), 20201234.

-L250. If so, our results may suggest a potential shift in synaptic plasticity accompanied mushroom body expansion, or that the increases in the amount of visual projection or calycal synapses affects this computation.

Is there any evidence for an increase of 'synaptic plasticity' variation across species' MB? If not, this hypothesis seems a bit far-fetched. I would rather stick to the evidence that MB variation is explain by a higher number of KCs (as demonstrated) and not a change in MB types of computation.

L 264 fwd. While further exploration of the cognitive shifts supported by mushroom body expansion are warranted, our data provide evidence that Heliconius have improved configurational learning, and more accurate long-term memory. These two functions are likely essential to meet the cognitive demands of learning the location of resources across space and time.

Again, it is not clear to me how an increase ability for configural learning (as tested with flower shape pattern recognition task) can support 'learning the location of resources across space and time'. There is evidence that such visual navigation, notably during trap-lining, is based on the recognition of egocentric scene (see Wystrach and Graham 2012) rather than the extraction and recombination of element required for configural learning (which is needed for flower recognition indeed). For instance, in Lihoreau et al., 2012. Traplining bumblebee would still visit the location of a flower, even if the flower has been removed, showing the use of scene rather than an acute recognition of flower.

Wystrach, A., & Graham, P. (2012). What can we learn from studies of insect navigation?. *Animal Behaviour*, 84(1), 13-20.

Lihoreau, M., Raine, N. E., Reynolds, A. M., Stelzer, R. J., Lim, K. S., Smith, A. D., ... & Chittka, L.

(2012). Radar tracking and motion-sensitive cameras on flowers reveal the development of pollinator multi-destination routes over large spatial scales.

L 283 by enhancements in aspects of cognitive performance which likely contribute to the cognitive toolkit required for these foraging innovations.

Given that it is unclear how these cognitive performances contribute to such foraging innovation (see point above), I would also mention the opposite scenario: that an increase in the MB size (selected to increase the memory capacity needed for memorising long visual route due to pollen feeding) provide a higher propensity to solve (and memorise) ambiguous visual pattern learning tasks in general.

-Figure 1. It would be useful to highlight which 5 species were chosen in B,C,D. (with an asterix in Fig 1a?)

-Finally, perhaps the work achieved in the same vein by Gronenberg across ants species should be cited?

Gronenberg, W., & Hölldobler, B. (1999). Morphologic representation of visual and antennal information in the ant brain. *Journal of Comparative Neurology*, 412(2), 229-240.

I hope my review has been helpful,

Reviewer #2 (Remarks to the Author):

Review: Couto et al. Rapid expansion

What accounts for the differences of brain organization across species and what do such differences mean? This question has challenged physicians and biologists long before 1859, with materialist studies that compared and contrasted mammalian brain organization by German Naturphilosophen, such as Carus and Oken, spreading to France and finally to Britain in the latter part of the 19th Century. From the very beginning, attention has focused on the organization of so-called "higher centers" of the brain, namely the cortex and observable differences of its folds and textures eventually leading to questions about neural types and variations of their stratification, mosaic organization, and any other geometrical relationship that met the eye. Only in the 1960's, with the advent of the ability to trace neuronal ensembles from sensory regions, such as the retina of olfactory bulb, has the interplay between sensory divergences and central neural architecture been of consuming interest. As the present authors opine, the assumption has developed that diversity and abundance of neural types shape brain composition and "provide the substrate for behavioral variation." At least, that is what many textbooks insist.

Scrutiny in depth as to whether this generalization is verifiable has been rare, with many superficial even anecdotal observations that show, for example, enlarged centers that relate to a specific modality occur in species for which that modality is of special relevance in behavior. Such circular rationale occupies too much literature without explaining how such differences have come about through natural selection, let alone what selective pressure came bear resulting in such diversity, a point rightly emphasized by the present authors in their Introduction. They also state, less convincingly, claims that diversity of neuronal types – thus logically diversity of connectivities – shape the organization of the brain. Yet it is not anomalous that comparisons of optic lobe organization in insect species, the variety of which offered Cajal and Sanchez a feast of comparative neuroanatomy, resolved multiple examples of morphologically divergent yet obviously

homologous neurons types, yet gave no hint whatsoever of the obvious differences of visual preferences and performance of, say, a tabanid and a honeybee.

Originally inspired by Dujardin's and Flögel's observations in the second half of the 19th Century ever more observers were drawn to volumes central to the insect brain, the paired mushroom bodies being the most attractive centers for speculation; from 1850, the very year of their discovery, they thenceforth carried the reputational baggage of being a pair of higher centers supporting intelligent behaviors, often exaggerated as the seat of insect cognition, what "cognition" implies for an insect is not just simple sensory associations and long term memory. In the last ten years the journal *eLife*, amongst others, has published dozens of highly sophisticated papers describing the extraordinary complexity of the fruit fly's mushroom body circuitry, its represented modalities, projections and associated centers (both efferent and afferent), as well as the repertoire of types of learning, memory and their modifications such as alteration and extinction. Referring to the Heliconiid mushroom body's size as suggesting multiple engram's suggests a rather uninformed perspective of what a mushroom body truly is. One fascinating finding, for example has been multimodal afferent information is represented by afferents to the mushroom body's lobes, rather than its calyces, the neuropil target of the present manuscript. A vast resource of information about the true cognitive capacity of a mushroom body is already available using the irrefragable deployment of genetic manipulation.

Genetic variation through natural selection fashions the mushroom body. Yet students of the *Drosophila* mushroom body have cared little if at all about how its organization came to be and the degree to which its mushroom bodies govern the fly's natural behavior at the levels of individual and population. Yet the organization of the fly's mushroom body is to a large degree generalist – multimodal – and thus representative of insect mushroom bodies in general. What is disconcerting, however, is that the fly mushroom body is minute, just 2,000 or so intrinsic neurons, with a circuit complexity that pales to insignificance when contrasted with its hymenopteran homologues. So yes; comparative studies which suggest differences of size and morphology clearly relate to behaviorally divergent behaviors that eventually have led to eusociality and truly collaborative cognition. In a word, mushroom bodies are a paradigm for high-functioning memory centers that relate to organize behaviors. It is because of this that the present manuscript offers entirely original and much needed insights as to how one obvious integer of complexity – mushroom body volume and thus its neuronal population -has evolved over time.

Considering all of the above, a research program that seeks to resolve the evolutionary history of mushroom body diversity is timely considering the many anecdotal studies that propose differences of mushroom body size can be ascribed to behavioral differences, such as navigational tasks. Early studies of mushroom bodies suggesting a role in allocentric memory have been reinvigorated by work showing that the vertical lobe of the mushroom body of ants is required for outbound navigation through feature detection. This behavioral negotiation has also been the subject of previous research by the lead author of the present study who showed that *Heliconius* butterflies employ trap-line foraging and collective roosting. Earlier studies, summarized in the manuscript, remind us that across a smaller sample of Heliconiid species, the particularly large volume of the mushroom body relates to that behavior.

The current manuscript describes the result of a comprehensive and elaborate program of research, the objective of which is to further underpin that finding by comparing two cardinal indicators. One is to show that across a range of Heliconiid species there has occurred in specific lineages differential evolution of mushroom body volume. The manuscript provides a huge data set that allows the mapping of mushroom traits onto the divergent evolutionary history of *Heliconius* sub-clades and their out-groups with the result that a lineage of pollen foragers that has also evolved extended reproductive longevity is endowed with mushroom bodies the volumes of which exceed those any of their relative species.

To reach this result, this team of researchers has had to pursue an arduous program of live animal collection and thence molecular genomic data providing relational trees that are anatomically independent of structural characters yet crucially information about genetic relationships. This has provide the scaffold on which anatomical traits are mapped to determine which lineages provide the observed anatomical expansion of the mushroom bodies, and in when (i.e. in geological time)

such expansion may have occurred. The answer is that enlargements of the mushroom body is due to an evolved increase in the number of its intrinsic neurons (Kenyon cells) and a concomitant increase in the density of synapses within the mushroom calyx, that part of the center that receives a defined visual input from the optic lobes and convergent input from the olfactory lobes. The crucial finding is that expansion of the mushroom body is a heritable trait that has evolved over a period of geological time that the authors estimate lies within 25 million years, hence more recent than any other proposed changes such as ascribed to the evolution of parasitoidism and hence eusociality in Hymenoptera.

The authors identify as one of the most crucial phylogenetic acquisitions enhanced visual discrimination and visual association memory. Their data also indicates that within the evolutionary history of the Heliconiids there has also been evolved loss in certain lineages, a finding that for this reviewer is an important indicator that their analysis resolves a true history of diversification: evolutionary story-telling that ignores evolved reduction and loss is uncomfortably unrealistic.

The manuscript itself is relatively short, with its concussively long supplementary data providing exhaustive descriptions of the research: from collecting to molecular sequencing and phylogenomics, neuroanatomy and quantitative analysis of cell numbers and synaptic densities. Herein lies a problem, which is that the density of description of the main narrative is in a language little different from that of the supplement. Thus a highly technical narrative that is likely incomprehensible to even the scientifically literate reader. Apart from an assertive introduction in layman's parlance the bulk of the manuscript is written in a language that is simply not accessible to a broader readership. And my prediction is that this publication will not resonate as it should unless the authors bite the bullet and seek assistance by a professional science writer than can render the technicalities into a language that serves to communicate the main points unobscured by what is (unfortunately) recognized as technobabble. The present reviewer, conversant in evolutionary analyses, neuroscience and animal cognition could not recommend this narrative to an aspiring graduate student. That's a pity, because the manuscript invites much deeper research, and much thought, particularly with regard to the cognitive abilities of insects, which in the present study are represented by rather simple assays that hardly evaluate aspects of deeper cognition. The manuscript also has a few difficulties that need to be addressed. One concern is that volumetric comparisons of mushroom bodies appear to be divorced from other higher centers and are made with respect to the volume of the standardized mid-brain. Yet we know that although mushroom bodies participate in feature recognition, and hence spatial exploration they do not support path integration. The fact that trap-line foraging species endowed with evolutionarily enlarged MBs undertake mass roosting, whereas other lineages do not, suggests that associative neuropils other than the mushroom bodies are likely more developed (*sensu lato*) than they are in species that do not use trap-line foraging. The authors need to considered this important aspect. One must take exception to statements that, in *Drosophila*, mushroom bodies are not essential for spatial memory. Such claims have been challenged by the relatively recent discovery of visual input to the fly's mushroom body. In general, spatial memory studies on flies have unknowingly (then but no longer) addressed path integration, not object recognition.

Claiming a link between neural elaboration and behavioral innovation is nothing new, and what should be emphasized more is that the present strategy described in this manuscript goes much further than any other in furthering Darwin's original proposal that heritable behavioral diversity is generated by ecological adaptation over geological time.

Finally, a few cautionary remarks. The authors would be well advised to lessen some of the hyperbole: such as "massive independent evolution..." Nobody will understand what "massive" evolution is. Also, "dramatic," "multiple bursts," are meaningless: if the authors are suggesting that "massive (!) increases" indicate rapid cladogenesis *sensu* Eldridge and Gould, then they should say so as it looks as if this underlies the describe mushroom body expansions. The occasional embarrassing slip up happens in most publications. It's worth catching them in manuscript mode. Thus, "configurational learning, particularly the integration of multisensory cues, plays a crucial role in insect navigation^{25,48} and these results..." Reference 48 describes homing in a whip spider not an insect.

RESPONSE TO REVIEWERS' COMMENTS

Reviewer 1

- *This work presents strong evidence for a link between the evolution of pollen feeding (at the base of the Heliconius genus) and mushroom bodies (MB) expansion. Further evidence shows convincingly that this expansion in MB size is due to a higher number of Kenyon Cell (rather than a change in MB structure), and correlates with an increased amount of visual projection to the MB. What's more, the authors demonstrate that this brain evolution is not a response to a change in the number of host plants, or the degree of social roosting, but is tightly link to the evolution pollen feeding which is known to involve trap-ling behaviours and thus the need for long route visual learning. Literature in ants and bees as shown indeed that long-route learning requires a large amount of KCs (receiving visual input), providing a very pleasant rationale to the story.*

The amount of data presented, as well as the work necessary to produce them is extremely consequent. The idea that MB evolved to support visual navigation was already in the literature for quite some time (see Webb and Wystrach 2016 for a review) mostly due to work in central place foraging hymenopteran; but the evidence was rather sparse. This work therefore provides the strongest support for this idea, and what's more, demonstrate it in an entirely different group of insects, and through a rapid radiation that is much more recent and with a clearly identified behavioural need (route following for pollen feeding in butterflies). I have no doubt that this study will stand as a hallmark of our understanding of the evolution of brains in insects and I therefore strongly recommend it for publication in Nature communication. This said, I have a single, major concern that could be easily alleviated.

Many thanks for these positive and encouraging comments.

- *Major point: There seem to be an apparent confusion in the ecological explanation provided. A confusion between, on the one hand the need for a large amount of Kenyon cells (and hence large MB) to increase memory space for visually learning long route; and on the other hand the general term of 'cognitive performance' and the ability to solve experimental tasks such as the one you tested here (ambiguous color-flower discrimination testing the notion of configural learning).*

It is true that both visual route following and experimental paradigm for configural learning, require the MB; however, it is really not clear whether the latter is required for the former (as implied here), or whether they have anything to do with each other (apart from requiring the MB). On the contrary, our current understanding of route following is that it requires the egocentric learning of wide visual field input (the whole scene) without extraction of its individual components. Large KC number are required to store more visual scene, and thus longer route. This is something quite different that the ability for fine and ambiguously coloured flower recognition (which may indeed be helped by larger KCs enabling a sparser coding, and thus affording more discrimination.

I see here therefore two different functions, one is about the evolution of memory space (for long route learning), and one is about the memorisation of fine visual discrimination (required to solve the paradigm presented here). I would thus recommend to present these as alternative hypothesis, rather than attempting to mingle them into a single one. One evolutionary scenario (which seem to be the one

backed up in the conclusion) is that 'high cognitive performance' is required for traplining (we don't have evidence for that); the other is that traplining require learning long routes and thus a high number of visual scene, which requires a large memory space (hence lots of KCs), and that large amount of KCs provide, as a byproduct, a better ability to solve experimental task requiring to discriminate and memorise ambiguous visual pattern. All details and references regarding these points are provided in the minor comments below.

Many thanks for these thoughts. We agree there was a lack of clarity over the context for these experiments. We have endeavoured to implement the changes suggested here, and in what follows, and detail more specific edits below.

- *"L51. This dietary innovation is accompanied by the evolution of trap-line foraging, where individuals learn foraging routes between resources with high spatial and temporal fidelity^{5,14,15}. Among insects, this foraging strategy is also found among species of bee^{16,17}, and is thought to rely on visual memories and landmark cues¹⁸."*

The first sentence is absolutely right, but the second is a bit confusing. It is not strictly true that trap-lining requires the learning of visual cues. There are models showing that this ability requires before all the ability to form celestial vector memory (involving rather the central complex, see LeMoel et al., 2019). However, it is indirectly true: Trap-lining requires the memorisation of long routes and multiple locations, and this, in turn (whether traplining or not) is known to require the memorisation of a large amount of visual scenes, as shown in ants (who don't trapline) and bees (who do trapline).

I think this subtlety is important to convey, as the argument that large amount of KCs is needed for learning visual scenes for navigation comes before all from studies in ants.

*Le Moël, F., Stone, T., Lihoreau, M., Wystrach, A., & Webb, B. (2019). The central complex as a potential substrate for vector based navigation. *Frontiers in psychology*, 10, 690.*

Thanks for this suggestion. We have updated the references and edited the sentence to read:

"Among insects, this foraging strategy is also found among some species of bee, and requires the ability to form vector memories^{16,17} and store large amount of visual scenes^{18,19}, potentially including landmark cues^{20,21}. (Lines 55-57).

- *"L 60 However, knowledge of how the adaptative evolution of sensory integration and learning centres supports cognitive enhancement is limited."*

Introduction jump between the rationale for the need for better 'spatial orientation' skills or 'cognitive enhancement'. It is unclear how this two terms relate in their rationale. What selective pressure is thought to have increase the MB (see major point).

We have removed his sentence and replaced it with the following text:

"However, the adaptive benefit of mushroom body expansion remains largely unestablished. Increased mushroom body size may facilitate increased memory space,

which is likely essential for the memorisation of multiple visual scenes to support learned foraging routes across large spatial scales³⁴. However, given their role in sensory integration and both elemental and more complex learning tasks^{36,37} larger mushroom bodies may also support more general cognitive enhancements through greater sensory discrimination and behavioural precision, through sparse coding of stimuli^{38,39}.” (Lines 66-73).

- “L58. While not essential for spatial memory in *Drosophila*²², mounting evidence from empirical^{23–25} and theoretical modelling^{26,27} implicates these structures in spatial orientation in other insects.”

Ref 27 (Baddeley et al., 2012) is not appropriate, as it is not a mushroom body model. However, Webb and Wystrach 2016 review the evidence for the role of large MB bodies in insect visual navigation.

Webb, B., & Wystrach, A. (2016). Neural mechanisms of insect navigation. *Current Opinion in Insect Science*, 15, 27-39.

We have removed ref 27 and replaced it with Webb & Wystrach 2016. We've also edited this sentence in response to a comment from reviewer 2 about spatial learning in *Drosophila*. It now reads:

“While mushroom bodies have previously been viewed as non-essential for spatial memory in *Drosophila*^{25,26}, more recent data do suggest a role in spatial memory²⁷ with more visual input to the calyx than previously appreciated²⁸. Combined with mounting evidence from empirical^{29–33} and theoretical modelling^{34,35} in other insects, these data strongly implicate the mushroom bodies in learnt spatial behaviours. (Lines 62-66).

- “L76 mushroom body volume is associated with variation in both visual and olfactory structures (medulla: $pMCMC = 0.002$; 77 antennal lobe: $pMCMC < 0.001$).”
I assume the correlation is positive? Would be good to make this clearer.

Indeed, both are positive, we have added this to the main text. (Lines 88-89).

- “L 84. Interestingly,⁸⁴ we also identify a *Heliconius* specific effect of sex (interaction *Heliconius**Sex $pMCMC < 0.001$),⁸⁵ with females having larger mushroom bodies on average than males ($pMCMC < 0.001$; Figure S2), which could reflect sex differences in foraging for pollen or host plant resources²⁹.”

Interesting indeed. I would however mention here what are these differences in foraging (from ref 29). Is the evidence in the good direction (with females foraging across more locations, or needing longer route) ?

Unfortunately, the field data is insufficient to answer this question precisely, but we have edited this sentence to read:

“Interestingly, we also identify a *Heliconius* specific effect of sex (interaction *Heliconius**Sex $pMCMC < 0.001$), with females having larger mushroom bodies on average than males ($pMCMC < 0.001$; Figure S2). Among wild caught *Heliconius* females tend to exhibit larger pollen loads⁴⁰, forage earlier in the day, and cover smaller areas compared to males, focusing on more local floral resources⁴¹. This possibly reflects less deviation from established

trap-lines. In some populations females may also use distinct pollen plants⁴².” (Lines 98-104).

- “L91 First, we used BAYOU30 to identify shifts in scaling between mushroom body size and central brain size.”

I am not qualified to assess the pertinence of the phylogenetic analysis (such as the use of BAYOU). However, it would be good to add a sentence to justify why the MB size was scaled relative to the central brain size (the latter being used as a control I assume) rather than the medulla, AL or overall brain size. But can this method disentangle between an evolutionary increase in MB size vs. a decrease in central brain size ?

It is concluded L103 ‘change in mushroom body size independently of central brain size. But, apart from the fact that the MB variation is greater than the central brain variation, it is not yet clear to me how the opposite hypothesis (change of central brain size independently of MB size is rejected based on this approach).

The subsequent phylogenetic analyses (rate based, L105 fwd) however seem to answer my above concern. My question is then: why not presenting the rate analyses first (based on absolute size), and then, as a second step, back up the results with the Bayou analysis (based on relative size) ?

The reviewer is indeed correct that these two analyses form a package that help to refine the inference. Controlling for allometric effects is a standard and essential component of comparative analyses of trait evolution. It is necessary to do so to show that there are specific, non-allometric shifts in trait size, a signature of adaptive evolution. We prefer to establish this first, before going on to show that these shifts are due to accelerated rates of evolution in the mushroom body specifically, rather than the allometric control which represents overall brain size.

What trait is best used as an allometric control is, of course, debatable. We prefer to use rest-of-CBR volume (rCBR) over whole brain size, for example, because it allows us to compare results for multiple brain components (MB, antennal lobe, medulla) without variation in those traits impacting the analysis of the others. In our experience rCBR is also less variable across species compared to other brain regions such as the medulla or antennal lobe, which are often under divergent selection pressures across species due to their sensory environment. rCBR is also highly predictive of the volume of other major neuropils (see for example, Figure 2E,F). As we are interested in controlling for general size effects, more variable traits likely introduce more “noise” in the analyses, complicating their interpretation.

We have added some clarifying text to the main manuscript (Lines 110-112 and 427-431).

- “L 168 To test this, we first confirmed that visual brain regions send projections to the mushroom body by injecting fluorescent retrograde tracers into the calyx.” It seems (Fig S8) that this analysis was conducted with the single species (*H. Melpomene*). I don’t think this is a problem, but it should be mentioned in the main text.

This is correct, we have clarified this in the text (Lines 202-203).

- “L218. One outstanding explanation is that the cognitive processes required for trap-line foraging are co-opted or refined from those supporting other, related, behaviors” I am not sure I understand this point. I believe this sentence is unclear because the concept behind is unclear. Why not simply suggesting that the need to learn longer routes and hence to form more visual memories of the scenes, be the reason? (see major point)

We have tried to clarify this sentence. Effectively the reviewer states what we meant, as our hypothesis is that territorial behaviour may require a degree of spatial fidelity supported by visual memories, and that this capacity is expanded in trap-lining species. The passage now read:

“One additional explanation for the independent increases in mushroom bodies in non-trap lining species, such as *Eueides* and *Dryadula*, is that they possess an ability to form visual memories as part of their wider behavioral repertoire. For example, *Heliconius* have strong site fidelity and homing ability, a likely pre-requisite for trap-lining, but also a trait shared with at least some territorial species of *Eueides*. True site fidelity likely requires a degree of spatial memory^{58,59}, and mushroom body expansion would therefore reflect an increase in the capacity to store more of these memories. We therefore suggest variation in this function may explain independent shifts in mushroom body size, and provide the foundation for the extreme expansion observed in *Heliconius*. Unfortunately, current ecological data on the movement ecology of non-*Heliconius* Heliconiini is limited, prohibiting formal tests of this hypothesis.” (Lines 256-266).

- “L244 Configural learning, particularly the integration of multisensory cues, plays a crucial role in insect navigation^{25,48}, and these results suggest that mushroom body expansion in *Heliconius* may support improvements in this ability.”

I am not convinced by the pertinence of this statement. First of all, ‘configural learning’ as tested here (with bicolour plants) is not a multisensory task. Second, I am not sure the evidence cited support the claim. Ref 25 (Bulehman et al., 2020 in ants) do not focus on configural learning at all; and Ref 48 (Flanigan et al., 2021, in whip spiders) concerns a discrimination task, which has not much to do with navigation (and even less insect navigation or visual navigation).

However, there is direct, albeit independent, evidence in insects that: on the one hand, visual navigation and notably long route following requires (and do correlates with) large amount of KCs (reviewed in Webb and Wystrach 2016); and on the other hand, that cognitive experimental paradigms such as the one you tested here (testing for the concept of configural learning) also requires the MB (Devault et al., 2015; Durrieu et al., 2020). However, the two type of tasks are quite different, and it is not clear how they converge ecologically (apart from the fact that they both require the MB).

More generally, it is not clear how the paradigm you used here with coloured flower-shaped pattern discrimination can relate directly to the ability to follow long routes. However, it demonstrates clearly a superior ability of the *Helicon*s species tested here to discriminate between ambiguous flowers using colour vision. Now the question is, does this relate to pollen feeding? And if yes, how ?

Devaud, J. M., Papouin, T., Carcaud, J., Sandoz, J. C., Grünewald, B., & Giurfa, M. (2015). Neural substrate for higher-order learning in an insect: mushroom bodies are necessary for configural discriminations. *Proceedings of the National Academy of Sciences*, 112(43), E5854-E5862.

Durrieu, M., Wystrach, A., Arrufat, P., Giurfa, M., & Isabel, G. (2020). Fruit flies can learn non-elemental olfactory discriminations. Proceedings of the Royal Society B, 287(1938), 20201234.

Thank you for these insights. We have edited the structure of the results to take this on board and to clarify our current hypotheses. We first discuss the hypothesised impact of increased Kenyon cell number in the context of memorising visual scenes at the end of section iii (*lines 185-196*). This more directly links the changes we observe to the current understanding of visual navigation in other insects.

We have then split the results from the cognitive experiments (formerly section vi) into two sections (vi and vii). The first presents the configural learning experiments (*starting line 268*). In the latter, we include some brief discussion on the potential use of more precise visual discrimination, illustrated by our configural learning assays, to support navigation via visual scene memorisation, while acknowledging the lack of data for the current system. We then also present the hypothesis suggested by the reviewer that the differences observed may reflect selection for traits not directly linked to trap-lining – for example we note that *Heliconius* do collect pollen from restricted range of floral plants so may have evolved increased discrimination for this purpose. The second deals with the long-term memory experiments, which we contextualise in terms of the increased individual longevity in *Heliconius* and the use of stable trap-lines across weeks or months (*starting line 303*).

We also no longer refer to multi-sensory tasks.

We hope these changes address the reviewer's comments sufficiently, and welcome any further suggestions for improvements.

- *“L250. If so, our results may suggest a potential shift in synaptic plasticity accompanied mushroom body expansion, or that the increases in the amount of visual projection or calycal synapses affects this computation.”*
Is there any evidence for an increase of ‘synaptic plasticity’ variation across species’ MB? If not, this hypothesis seems a bit far-fetched. I would rather stick to the evidence that MB variation is explain by a higher number of KCs (as demonstrated) and not a change in MB types of computation.

We do have data which we believe shows this effect, but it is currently unpublished. We note, however, that there is published data available which illustrates variable mushroom body plasticity in other butterflies (van Dijk et al 2017, Proc Roy Soc B). We have edited this sentence to read:

“Our results suggest that increases in the amount of visual projection neurons to the mushroom body calyx in *Heliconius*, and the associated greater number of Kenyon cells and synapses may also affect this computation.” (*Lines 290-292*).

- *“L 264 fwd. While further exploration of the cognitive shifts supported by mushroom body expansion are warranted, our data provide evidence that Heliconius have improved configurational learning, and more accurate long-term memory. These two functions are likely essential to meet the cognitive demands of learning the location of resources across space and time.”*

Again, it is not clear to me how an increase ability for configural learning (as tested with flower shape pattern recognition task) can support 'learning the location of resources across space and time'. There is evidence that such visual navigation, notably during trap-lining, is based on the recognition of egocentric scene (see Wystrach and Graham 2012) rather than the extraction and recombination of element required for configural learning (which is needed for flower recognition indeed). For instance, in Lihoreau et al., 2012. Traplining bumblebee would still visit the location of a flower, even if the flower has been removed, showing the use of scene rather than an acute recognition of flower.

*Wystrach, A., & Graham, P. (2012). What can we learn from studies of insect navigation?. *Animal Behaviour*, 84(1), 13-20.*

Lihoreau, M., Raine, N. E., Reynolds, A. M., Stelzer, R. J., Lim, K. S., Smith, A. D., ... & Chittka, L. (2012). Radar tracking and motion-sensitive cameras on flowers reveal the development of pollinator multi-destination routes over large spatial scales.

We appreciate the point and have deleted the second of these sentences as part of our restructuring of this section of the manuscript.

As an aside, there is also field data suggesting *Heliconius* continue to visit locations of pollen resources that have been removed, but update their trap-line over time (Ehlich and Gilbert 1973, *Biotropica*, 69-82).

- *“L 283 by enhancements in aspects of cognitive performance which likely contribute to the cognitive toolkit required for these foraging innovations.”*
Given that it is unclear how these cognitive performances contribute to such foraging innovation (see point above), I would also mention the opposite scenario: that an increase in the MB size (selected to increase the memory capacity needed for memorising long visual route due to pollen feeding) provide a higher propensity to solve (and memorise) ambiguous visual pattern learning tasks in general.

We have edited this sentence to more directly reflect the results. It now reads:

“However, we provide evidence that the most substantial shifts in size and sensory modality co-occur with the origin of pollen feeding and trap-line foraging. These neuroanatomical shifts were accompanied by enhancements in long term visual memory, and a capacity to discriminate more complex visual patterns. We hypothesise that these behavioural differences reflect either a direct response to selection on foraging behavior, correlated traits such as increased longevity or floral specificity, or an indirect consequence of increased Kenyon cell number.” (Lines 335-341).

- *“Figure 1. It would be useful to highlight which 5 species were chosen in B,C,D. (with an asterix in Fig 1a?)”*

Thanks for this suggestion, we have edited the figure as suggested to better link panel A to panels B,C and D.

- *“Finally, perhaps the work achieved in the same vein by Gronenberg across ants species should be cited?”*

Gronenberg, W., & Hölldobler, B. (1999). Morphologic representation of visual and antennal information in the ant brain. *Journal of Comparative Neurology*, 412(2), 229-240.”

Agreed, we now include this among the references.

□ *I hope my review has been helpful,*

Very much so, we are grateful for your thoughts and constructive comments which have helped clarify and improve the manuscript, and welcome any further suggestions you might have on the revised documents.

Reviewer 2

□ *What accounts for the differences of brain organization across species and what do such differences mean? This question has challenged physicians and biologists long before 1859, with materialist studies that compared and contrasted mammalian brain organization by German Naturphilosophen, such as Carus and Oken, spreading to France and finally to Britain in the latter part of the 19th Century. From the very beginning, attention has focused on the organization of so-called “higher centers” of the brain, namely the cortex and observable differences of its folds and textures eventually leading to questions about neural types and variations of their stratification, mosaic organization, and any other geometrical relationship that met the eye. Only in the 1960’s, with the advent of the ability to trace neuronal ensembles from sensory regions, such as the retina of olfactory bulb, has the interplay between sensory divergences and central neural architecture been of consuming interest. As the present authors opine, the assumption has developed that diversity and abundance of neural types shape brain composition and “provide the substrate for behavioral variation.” At least, that is what many textbooks insist.*

Scrutiny in depth as to whether this generalization is verifiable has been rare, with many superficial even anecdotal observations that show, for example, enlarged centers that relate to a specific modality occur in species for which that modality is of special relevance in behavior. Such circular rationale occupies too much literature without explaining how such differences have come about through natural selection, let alone what selective pressure came bear resulting in such diversity, a point rightly emphasized by the present authors in their Introduction. They also state, less convincingly, claims that diversity of neuronal types – thus logically diversity of connectivities – shape the organization of the brain. Yet it is not anomalous that comparisons of optic lobe organization in insect species, the variety of which offered Cajal and Sanchez a feast of comparative neuroanatomy, resolved multiple examples of morphologically divergent yet obviously homologous neurons types, yet gave no hint whatsoever of the obvious differences of visual preferences and performance of, say, a tabanid and a honeybee.

Originally inspired by Dujardin’s and Flögel’s observations in the second half of the 19th Century ever more observers were drawn to volumes central to the insect brain, the paired mushroom bodies being the most attractive centers for speculation; from 1850, the very year of their discovery, they thenceforth carried the reputational baggage of being a pair of higher centers supporting intelligent behaviors, often exaggerated as the seat of insect cognition, what “cognition” implies for an insect is

not just simple sensory associations and long term memory. In the last ten years the journal eLife, amongst others, has published dozens of highly sophisticated papers describing the extraordinary complexity of the fruit fly's mushroom body circuitry, its represented modalities, projections and associated centers (both efferent and afferent), as well as the repertoire of types of learning, memory and their modifications such as alteration and extinction. Referring to the Heliconiid mushroom body's size as suggesting multiple engram's suggests a rather uninformed perspective of what a mushroom body truly is. One fascinating finding, for example has been multimodal afferent information is represented by afferents to the mushroom body's lobes, rather than its calyces, the neuropil target of the present manuscript. A vast resource of information about the true cognitive capacity of a mushroom body is already available using the irreproachable deployment of genetic manipulation.

Genetic variation through natural selection fashions the mushroom body. Yet students of the Drosophila mushroom body have cared little if at all about how its organization came to be and the degree to which its mushroom bodies govern the fly's natural behavior at the levels of individual and population. Yet the organization of the fly's mushroom body is to a large degree generalist – multimodal – and thus representative of insect mushroom bodies in general. What is disconcerting, however, is that the fly mushroom body is minute, just 2,000 or so intrinsic neurons, with a circuit complexity that pales to insignificance when contrasted with its hymenopteran homologues. So yes; comparative studies which suggest differences of size and morphology clearly relate to behaviorally divergent behaviors that eventually have lead to eusociality and truly collaborative cognition. In a word, mushroom bodies are a paradigm for high-functioning memory centers that relate to organize behaviors. It is because of this that the present manuscript offers entirely original and much needed insights as to how one obvious integer of complexity – mushroom body volume and thus its neuronal population -has evolved over time.

Many thanks for this enjoyable description of the context of the paper!

- *Considering all of the above, a research program that seeks to resolve the evolutionary history of mushroom body diversity is timely considering the many anecdotal studies that propose differences of mushroom body size can be ascribed to behavioral differences, such as navigational tasks. Early studies of mushroom bodies suggesting a role in allocentric memory have been reinvigorated by work showing that the vertical lobe of the mushroom body of ants is required for outbound navigation through feature detection. This behavioral negotiation has also been the subject of previous research by the lead author of the present study who showed that Heliconius butterflies employ trap-line foraging and collective roosting. Earlier studies, summarized in the manuscript, remind us that across a smaller sample of Heliconiid species, the particularly large volume of the mushroom body relates to that behavior.*

The current manuscript describes the result of a comprehensive and elaborate program of research, the objective of which is to further underpin that finding by comparing two cardinal indicators. One is to show that across a range of Heliconiid species there has occurred in specific lineages differential evolution of mushroom body volume. The manuscript provides a huge data set that allows the mapping of mushroom traits onto the divergent evolutionary history of Heliconius sub-clades and their out-groups with the result that a lineage of pollen foragers that has also evolved

extended reproductive longevity is endowed with mushroom bodies the volumes of which exceed those any of their relative species.

To reach this result, this team of researchers has had to pursue an arduous program of live animal collection and thence molecular genomic data providing relational trees that are anatomically independent of structural characters yet crucially information about genetic relationships. This has provide the scaffold on which anatomical traits are mapped to determine which lineages provide the observed anatomical expansion of the mushroom bodies, and in when (i.e. in geological time) such expansion may have occurred. The answer is that enlargements of the mushroom body is due to an evolved increase in the number of its intrinsic neurons (Kenyon cells) and a concomitant increase in the density of synapses within the mushroom calyx, that part of the center that receives a defined visual input from the optic lobes and convergent input from the olfactory lobes. The crucial finding is that expansion of the mushroom body is a heritable trait that has evolved over a period of geological time that the authors estimate lies within 25 million years, hence more recent than any other proposed changes such as ascribed to the evolution of parasitoidism and hence eusociality in Hymenoptera.

The authors identify as one of the most crucial phylogenetic acquisitions enhanced visual discrimination and visual association memory. Their data also indicates that within the evolutionary history of the Heliconiids there has also been evolved loss in certain lineages, a finding that for this reviewer is an important indicator that their analysis resolves a true history of diversification: evolutionary story-telling that ignores evolved reduction and loss is uncomfortably unrealistic.

Thanks again for this summary, it is gratifying to see that the amount of work involved has been recognised.

- *The manuscript itself is relatively short, with its concussively long supplementary data providing exhaustive descriptions of the research: from collecting to molecular sequencing and phylogenomics, neuroanatomy and quantitative analysis of cell numbers and synaptic densities. Herein lies a problem, which is that the density of description of the main narrative is in a language little different from that of the supplement. Thus a highly technical narrative that is likely incomprehensible to even the scientifically literate reader. Apart from an assertive introduction in layman's parlance the bulk of the manuscript is written in a language that is simply not accessible to a broader readership. And my prediction is that this publication will not resonate as it should unless the authors bite the bullet and seek assistance by a professional science writer than can render the technicalities into a language that serves to communicate the main points unobscured by what is (unfortunately) recognized as technobabble. The present reviewer, conversant in evolutionary analyses, neuroscience and animal cognition could not recommend this narrative to an aspiring graduate student. That's a pity, because the manuscript invites much deeper research, and much thought, particularly with regard to the cognitive abilities of insects, which in the present study are represented by rather simple assays that hardly evaluate aspects of deeper cognition.*

We are naturally disappointed the reviewer did not find the writing as clear as it could be, we have had positive feedback from other readers but appreciate the point raised here. We have gone through the manuscript to edit any passages which are too dense and expand on more technical aspects of the work. We'd be grateful for any further thoughts on specific areas which remain unclear.

- The manuscript also has a few difficulties that need to be addressed. One concern is that volumetric comparisons of mushroom bodies appear to be divorced from other higher centers and are made with respect to the volume of the standardized mid-brain. Yet we know that although mushroom bodies participate in feature recognition, and hence spatial exploration they do not support path integration. The fact that trap-line foraging species endowed with evolutionarily enlarged MBs undertake mass roosting, whereas other lineages do not, suggests that associative neuropils other than the mushroom bodies are likely more developed (*sensu lato*) than they are in species that do not use trap-line foraging. The authors need to consider this important aspect.

We of course agree, the mushroom body does not operate in isolation. We include analyses of the medulla and antennal lobe to consider the possibility that the shifts observed reflect changes upstream of the mushroom bodies themselves, but we appreciate there may also be changes elsewhere in the brain. In particular, we imagine the reviewer is thinking of the central complex, which has established roles in navigation and spatial orientation. We have indeed been considering changes in these neuropils. For the reviewer's interest, we find no evidence that the central complex differs significantly in volume between *Heliconius* and other Heliconiini. This can, for example be seen in a plot of the central body volumes:

You can see that the data for the two groups overlap, indicating a lack on a pollen-feeding associated shift in central complex size. As a result, when central complex volume is plotted against mushroom body size, in panel B, there is a separation between *Heliconius* and other genera, with *Eueides* and *Dryadula* intermediate between the *Heliconius* and other genera. This basically mirrors the pattern shown in Figure 2B when mushroom body volume is plotted against rCBR.

We prefer not to include this here as we are also currently investigating more fine-grained differences in neurotransmitter expression, and internal morphology in more detail. As it is possible more subtle changes in central complex anatomy or neurochemistry accompany mushroom body expansion. This data collection is ongoing, and we plan to present it as a package with the volumetric data in the near future.

In the current manuscript, we therefore now allude to the possibility raised by the reviewer (*Lines 137-139*).

- One must take exception to statements that, in *Drosophila*, mushroom bodies are not essential for spatial memory. Such claims have been challenged by the relatively

recent discovery of visual input to the fly's mushroom body. In general, spatial memory studies on flies have unknowingly (then but no longer) addressed path integration, not object recognition.

We agree this was an oversimplification. We have edited this sentence to now read:

*"While mushroom bodies have previously been viewed as non-essential for spatial memory in *Drosophila*^{25,26}, more recent data do suggest a role in spatial memory²⁷ with more visual input to the calyx than previously appreciated²⁸. Combined with mounting evidence from empirical²⁹⁻³³ and theoretical modelling^{34,35} in other insects, these data strongly implicate the mushroom bodies in learnt spatial behaviours."
(Lines 62-66).*

- *Claiming a link between neural elaboration and behavioral innovation is nothing new, and what should be emphasized more is that the present strategy described in this manuscript goes much further than any other in furthering Darwin's original proposal that heritable behavioral diversity is generated by ecological adaptation over geological time.*

We appreciate this comment and have edited the conclusion to stress this point (Lines 341-345).

- *Finally, a few cautionary remarks. The authors would be well advised to lessen some of the hyperbole: such as "massive independent evolution..." Nobody will understand what "massive" evolution is. Also, "dramatic," "multiple bursts," are meaningless: if the authors are suggesting that "massive (!) increases" indicate rapid cladogenesis sensu Eldridge and Gould, then they should say so as it looks as if this underlies the describe mushroom body expansions.*

We have edited the manuscript to more directly and quantitatively reflect the results in these instances.

- *The occasional embarrassing slip up happens in most publications. It's worth catching them in manuscript mode. Thus, "configurational learning, particularly the integration of multisensory cues, plays a crucial role in insect navigation^{25,48} and these results..." Reference 48 describes homing in a whip spider not an insect.*

Indeed! Many thanks for spotting this slip.

REVIEWERS' COMMENTS

Reviewer #1 (Remarks to the Author):

The authors did a great job at implementing my comments and suggestions. Most importantly, the apparent entanglement between hypotheses (cognitive flexibility, memory capacity and their relation to trap-lining) and insight from the data (discrimination, long term memory) is now lifted. The suggestion to relay the BAYOU analysis as a second step was only a suggestion indeed, and the response of the authors convinced me that it is better to keep things as they are regarding this point. My concern about the lack of evidence for change in synaptic plasticity in the MB was equally answered, I was not aware of this work in butterflies. Overall, the manuscript feels now much sharper, the hypotheses themselves, as well as their relation to the current and previous data is now clear and sound. I have no further recommendation.

Excellent work,
With best wishes,

Reviewer #2 (Remarks to the Author):

I thank the authors for revising the narrative. It's easier to read, mainly because it flows better. The other relative minor comments are resolved as is the major one concerning the now established role of mushroom bodies in place memory (glad that the authors admit fruitflies to the fold). One major concern, which isn't resolved and apparently isn't seen as crucial, is the vexing matter of what has driven the evolution, over a very short period of geological time, of Kenyon cell proliferation. The calyces aren't demonstrably supplied by massive visual inputs, at least not to the extent that occurs in many hymenopterans. Yet the authors show convincingly that visual memory is a property of the mushroom body lobes, i.e. the multitudinous synaptic interactions of K-cell parallel fibers (these are not axons by the way: K-cells cannot be treated as input-output relay neurons, but as local interneurons). So where do they receive high level visual inputs? Isn't the answer likely to be found in the lobes, the multimodalities of which are ignored in so many publications, including the present manuscript? It has been demonstrated for several species that that the MB lobes receive multisensory afferents. Isn't this likely for lepidopterans as well? It's certainly true of stemward Odonata, for example, which entirely lack calyces, have massive mushroom body lobes receiving a rich supply of afferents, and which have wonderfully fluid place memory. Blattodea provide another example in which visual supply to the mushroom bodies is exclusive to the lobes.

I'm not going to insist that the authors revise this paper to take those considerations into account, because there is no evidence that they have data on the internal organization of the lobes. Rather, in the present manuscript these are considered solely as storage devices and output levels. Yet, assuming they exist, visual/multimodal inputs to the lobes would be as crucial heritable traits of the mushroom bodies as is its calycal supply. So I find it disconcerting that this is left out of consideration. The same discomfort is provided by the Kinoshita et al. 2014 paper on *Papilio xuthus*. It's understandable that additional complexity might threaten to muddy the waters: the more simplistic a mushroom body, the easier it is to deal with it as a Hebbian device receiving input at one specialized level only. However, the calyces cannot be assumed as symplesiomorphies, at least if one accepts current insect molecular phylogenetics.

So, with that concern I will leave this manuscript alone. Its paucity of mushroom body neuroanatomy is, for those that consider it important, extremely concerning: down the road the publication will be exposed to criticism. I urge that future work on this system looks closely at what's going beneath the surface, meaning all the subsequent levels of integration in the lobes. The many papers on the fly mushroom in eLife should allow several handy leaves for the author's further perusal.

RESPONSE TO REVIEWERS' COMMENTS

Reviewer 1

- The authors did a great job at implementing my comments and suggestions. Most importantly, the apparent entanglement between hypotheses (cognitive flexibility, memory capacity and their relation to trap-lining) and insight from the data (discrimination, long term memory) is now aligned. The suggestion to relay the BAYOU analysis as a second step was only a suggestion indeed, and the response of the authors convinced me that it is better to keep things as they are regarding this point. My concern about the lack of evidence for change in synaptic plasticity in the MB was equally answered, I was not aware of this work in butterflies. Overall, the manuscript feels now much sharper, the hypotheses themselves, as well as their relation to the current and previous data is now clear and sound. I have no further recommendation.*

We are very grateful for the reviewer's comments and are glad they are satisfied with the revisions.

No further action taken.

Reviewer 2

- I thank the authors for revising the narrative. It's easier to read, mainly because it flows better. The other relative minor comments are resolved as is the major one concerning the now established role of mushroom bodies in place memory (glad that the authors admit fruitflies to the fold). One major concern, which isn't resolved and apparently isn't seen as crucial, is the vexing matter of what has driven the evolution, over a very short period of geological time, of Kenyon cell proliferation. The calyces aren't demonstrably supplied by massive visual inputs, at least not to the extent that occurs in many hymenopterans. Yet the authors show convincingly that visual memory is a property of the mushroom body lobes, i.e. the multitudinous synaptic interactions of K-cell parallel fibers (these are not axons by the way: K-cells cannot be treated as input-output relay neurons, but as local interneurons). So where do they receive high level visual inputs? Isn't the answer likely to be found in the lobes, the multimodalities of which are ignored in so many publications, including the present manuscript? It has been demonstrated for several species that that the MB lobes receive multisensory afferents. Isn't this likely for lepidopterans as well? It's certainly true of stemward Odonata, for example, which entirely lack calyces, have massive mushroom body lobes receiving a rich supply of afferents, and which have wonderfully fluid place memory. Blattodea provide another example in which visual supply to the mushroom bodies is exclusive to the lobes.*

We are glad the reviewer feels the flow of the text is improved. We must admit some confusion regarding the evidence of visual input, which is dealt with in the main text and Figure 4, and was not a point raised previously by the reviewer. Nonetheless, we have reviewed the images from our tracing experiments and do not see any evidence of direct input from the optic lobes to the mushroom body lobes, although it is possible we do miss some small tracts.

- *I'm not going to insist that the authors revise this paper to take those considerations into account, because there is no evidence that they have data on the internal organization of the lobes. Rather, in the present manuscript these are considered solely as storage devices and output levels. Yet, assuming they exist, visual/multimodal inputs to the lobes would be as crucial heritable traits of the mushroom bodies as is its calycal supply. So I find it disconcerting that this is left out of consideration. The same discomfort is provided by the Kinoshita et al. 2014 paper on *Papilio xuthus*. It's understandable that additional complexity might threaten to muddy the waters: the more simplistic a mushroom body, the easier it is to deal with it as a Hebbian device receiving input at one specialized level only. However, the calyces cannot be assumed as symplexiomorphies, at least if one accepts current insect molecular phylogenetics.*

We do include an analysis of scaling between the lobes and calyx (lines 178-184), and are aware of the importance of its internal composition and connectivity. Indeed, similarly to our previous response regarding the central complex, we have an ongoing project investigating exactly this question. As can be seen in Figure 1, the compressed structure of the lobes makes this more challenging than in some other insects where distinct lobes are easily delineated. As such it has required a separate and time consuming approach, and is beyond the main focus of the current study which, as the reviewer acknowledges, already includes a huge amount of data and work.

- *So, with that concern I will leave this manuscript alone. Its paucity of mushroom body neuroanatomy is, for those that consider it important, extremely concerning: down the road the publication will be exposed to criticism. I urge that future work on this system looks closely at what's going beneath the surface, meaning all the subsequent levels of integration in the lobes. The many papers on the fly mushroom in eLife should allow several handy leaves for the author's further perusal.*

Indeed, we already cite the eLife paper. However, we struggle to reconcile this comment with the reviewer's previous comment describing the study as "a comprehensive and elaborate program of research". As noted above, and in our previous response, our work on this system is ongoing, multifaceted and will naturally develop across more than one publication.

No further action taken.